# Interactions between attributions and beliefs at trial-by-trial level: Evidence from a novel computer game task

**Elena Zamfir**[1]⊕*, **Peter Dayan**[2]⊕

**1** Department of Education, University of Oxford, Oxford, United Kingdom, **2** Max Plack Institute for Biological Cybernetics, Tuebingen, Germany

⊕ These authors contributed equally to this work.
* elena.zamfir@education.ox.ac.uk

**Data Availability Statement:** The source code and data used to produce the results and analyses presented in this manuscript are available at https://osf.io/sdvf5/.

## Abstract

Inferring causes of the good and bad events that we experience is part of the process of building models of our own capabilities and of the world around us. Making such inferences can be difficult because of complex reciprocal relationships between attributions of the causes of particular events, and beliefs about the capabilities and skills that influence our role in bringing them about. Abnormal causal attributions have long been studied in connection with psychiatric disorders, notably depression and paranoia; however, the mechanisms behind attributional inferences and the way they can go awry are not fully understood. We administered a novel, challenging, game of skill to a substantial population of healthy online participants, and collected trial-by-trial time series of both their beliefs about skill and attributions about the causes of the success and failure of real experienced outcomes. We found reciprocal relationships that provide empirical confirmation of the attribution-self representation cycle theory. This highlights the dynamic nature of the processes involved in attribution, and validates a framework for developing and testing computational accounts of attribution-belief interactions.

## Author summary

As part of interpreting our experiences, we spontaneously make causal attributions and use them to update our beliefs about the world, ourselves and others. This has long been a topic of interest, particularly within psychiatry. Some theories assume that people have stable "attributional styles", others focus on the changing nature of attribution-making and on the relationships between attributions and one's beliefs about the self, suggesting that the two are mutually connected. In this area of research, people have traditionally been asked to imagine themselves experiencing various significant life events and report on how they would interpret those, or have been exposed to artificial and highly simplified situations in the lab. In this work, we introduce a new task to study relationships between causal attributions and beliefs: repeatedly playing an engaging and relatively complex game of skill. We show that we can detect mutual influences between attributions and

**Funding:** The authors received no specific funding for this work.

**Competing interests:** The authors have declared that no competing interests exist.

beliefs at the level of individual wins and losses. This has implications for how everyday successes and failures impact our beliefs about ourselves and our well-being. It also could help understand how our interpretations of negative experiences can spiral out of control, affecting our mental health.

## Introduction

When people succeed or fail at achieving their goals, they typically use beliefs about their *abilities* or skills to *attribute* the result to their own internal efforts or to external factors. Conversely, people typically use appraisals of past successes and failures to inform their beliefs about themselves [1, 2].

A simple concrete example can be used to illustrate these two processes at work: consider a student who finds out they got a low mark in a math assignment. If the student has a strong belief in their mathematical ability or general intelligence they might assign this low grade to the assignment being excessively difficult or confusing—an *external* cause. Alternatively, they might consider themselves to have been tired and unable to properly focus when working on their homework—an *internal*, but *transient* cause. In contrast, a student lacking confidence in their math abilities might interpret the low grade as a result of their stupidity, and attribute the failure to this *internal* and *stable* factor—if one is stupid, one is likely to remain so in the future. Thus beliefs can influence causal attributions. Conversely, attributions can influence beliefs: assigning the low grade to external factors or to tiredness will most likely not affect the student's estimation of their ability in math; however attributing the low grade to stupidity is likely to strengthen their belief in their inability, and impact their motivation for studying this topic in the future.

Crucially, such appraisals do not occur in isolation: students will get a large number of various grades on their assignments or exams throughout their studies, creating the potential for interactions between successive such appraisals. Most students will successfully manage the emotional impact of such micro triumphs or disasters, maintaining their well-being while learning about and striving to improve their knowledge and skill base. However more extreme or even tragic path dependencies [3] are also possible: if a student who is slightly prone to low self-esteem suffers from a string of such failures, each could amplify the effect of the next, by strengthening the student's belief in their inadequacies, making them more likely to interpret feedback in a negative way and therefore further erode their self esteem. Getting trapped in such vicious cycles could drain the students' energy and motivation, making it increasingly difficult to maintain a healthy and productive approach to their studies, and affecting their mental health.

This example illustrates how pervasive such inferences are in everyday situations, but also shows the potential for reciprocal connections between causal attributions and beliefs at the level of individual events. If such interactions at high temporal resolution are indeed present, the accumulation of their effects could produce rich dynamics, ranging from the maintenance of a homeostatic regime to vicious cycles spiraling out of control [1, 2].

Abnormal beliefs and attributional patterns have been topics of particular interest in connection with psychiatric disorders, and a wealth of theoretical proposals has been put forth about the psychological mechanisms involved, and how they malfunction in various disorders. Early theories explaining the role of attributions in the generation and maintenance of depression postulated the existence of stable, trait-like "attributional styles" (the reformulated learned helplessness theory of depression; [4]) or "cognitive styles"—propensities for making specific

attributions for events and inferences about their consequences and the self (the hopelessness theory of depression; [5]) and a "feed-forward" causal chain model linking negative events to symptoms of depression.

Much of the research inspired by the theories of depression associated with learned helplessness [6–8] measures attributions through questionnaires involving hypothetical meaningful life events, a paradigmatic example being the Attributional Style Questionnaire (ASQ [9]). In cases where real outcomes were used, the typical experimental design involved either groups differing based on Beck Depression Inventory (BDI) scores or other measures of depression, or groups subjected to mood inducing manipulations ([10–12]), who were asked to perform a task and then report their attribution for the outcome. In these paradigms, attributions were often measured as contingency judgements [10, 13–20]. That is, participants estimated the degree of control they exercised over the appearance of stimuli in simple lab-based tasks involving button presses and contingent or non-contingent presentation of visual stimuli. Less often, the effect of attributions was measured as participants' evaluations of their own performance, or expectations of future performance, in tasks framed as either skill or chance tasks [21, 22], in an attempt to manipulate participants' attributions of the outcomes they experienced. Research involving relationships between attributional styles and beliefs about the self has found that measures of self-esteem can account for variation in attributional styles in both the general population and in psychiatric patients [23–25]. Notably, in accordance with the underlying assumptions about trait-like attributional styles, attributions were manipulated and/or measured at the *condition level*, producing aggregate rather than trial-by-trial measures.

More recently, the dynamic nature of the processes involved was brought into focus [1, 2]. In [1], Bentall et al. postulate that rather than having a trait-like attributional style, individuals use current beliefs about the self, or readily available stored knowledge about the self, along with attributional signposts in situational information [26] when making causal attributions. According to this theory, the process of attribution formation involves a search for explanations that starts with explanations involving the self and terminates when a suitable explanation is found; on the other hand, once an attribution is formed, it primes representations of the self that are consistent with it. Thus, along with effects of attributions on beliefs about the self, Bentall et al. recognise the possibility of effects in the opposite direction, leading to a system with fluctuating components and potentially complex effects of interactions between them—the attribution-self-representation cycle (ASRC) [2]. The system can be influenced by relatively stable factors, such as individual differences in stored knowledge about the self, motivational biases, tendency to attend to specific types of information or ability to understand others, as well as fluctuating factors determining the relative availability of information in different circumstances.

Within this framework, the self-serving or self-enhancing attributional biases that healthy people display ([13–15, 27–29]; for reviews, see [21, 30, 31]) function as homeostatic mechanisms for the maintenance of beliefs about the self within healthy parameters; the absence or disruption of these homeostatic mechanisms leaves people vulnerable to aberrant protective mechanisms or vicious cycles (where negative internal attributions lead to a worsening of self-beliefs, leading to further negative attributions), producing and or maintaining depressive or paranoid symptoms. The dynamic nature of both attributions and self-beliefs, and the fact that they exert reciprocal influences on each other can be expected to produce complex patterns of relationships between them [32]. In turn, the inherent difficulty of predicting such relationships could account for some of the inconclusive or contradictory findings of studies aimed at testing previous theories, whose predictive power was duly low (for reviews, see [33, 34]).

Research investigating the effect of manipulating experience on subsequent attributions produced evidence that this phenomenon is indeed more dynamic than previously considered [28, 35, 36]. In a series of experiments in which participants' mood was manipulated (either

through false feedback in an experimental task or through exposure to emotionally charged short films), [35] measured attributions made by healthy participants either for hypothetical situations or for their own real exam results and found that participants given a positive mood made more internal and stable attributions for positive outcomes, and less internal and stable attributions for negative outcomes, than participants given negative mood. Similarly, in a study involving clinical populations, [28] administered the expanded ASQ questionnaire to depressed, manic and normal participants and asked them to judge the contingency between their actions and outcomes in a computer-based task before and after exposing them to a mild failure experience (an anagram solving task which included unsolvable items). Negative internality scores increased after the failure experience for both groups of patients (although not for the healthy controls, who might be less vulnerable to the effects of experiencing such mild failures). Conversely, situations involving potential threats to self-esteem promoted self-serving attributions [36].

These studies showed that attribution-making tendencies can vary, *at the task timescale*, within the same population, and that factors such as mood or achievement can exert an effect on them. These results highlight the need for more precise investigations, specifically designed to uncover the dynamics of attributions and beliefs about the self at high temporal resolution. This would allow for a quantitative test of theoretical accounts based on time-varying interactions between these variables, such as the ASRC [1, 2, 28]. To this end, we designed and administered a novel engaging task of skill that healthy participants learned to perform whilst also providing us with time series of both attributions and beliefs about skill (skill estimates). This procedure differs from its predecessors in three key ways. First, participants experienced a series of real successes and failures as they learned. Second, we collected time series of *participants' own* attributions for outcomes and skill estimates at a fine, trial-level, temporal granularity. Third: we did not manipulate the content of participants' attributions or skill estimates. Importantly, although participants' responses were relatively unconstrained, their experience in the task was influenced by measurable and controllable external parameters, and their performance could be be objectively quantified. Finally, we aimed to make the task engaging enough for participants to care about the outcomes and their progress, and thus report meaningful attributions and beliefs about the self.

We found evidence of trial-level reciprocal effects of attributions and beliefs which could support the dynamics postulated by the ASRC: participants updated their skill estimates more after outcomes attributed internally than after ones attributed externally, and internal attributions for wins and losses varied with participants' previous skill estimates—they took more credit for wins and less blame for losses with increasing skill estimates. Moreover, individual differences in these effects correlated with traditional questionnaire-based measures of self esteem, locus of control and attributional style.

The rest of this paper is organised as follows: we begin by presenting simulations which introduce the formalism we will use throughout and illustrate the richness of dynamics that can be supported by trial-level reciprocal attribution-beliefs interactions; we then present the task and experimental design; we report the results of our analyses of the real data, focusing on skill estimates, attributions, and analyses of questionnaire responses. Finally, we reflect on the task and discuss our results, directions for future work and conclusions.

## Results

### Simulations of an artificial agent

In this section we present simulations of simple artificial agents endowed with evolving beliefs about skill and a mechanism for making causal attributions. We introduce our formalisation

of these variables and their interactions, and illustrate some of the behavioural patterns supported by reciprocal trial-by-trial effects between them. The point is not to provide a detailed model of the empirical investigation that follows in the remaining sections of the paper, but rather to provide a formalized underpinning for the interpretation of that experiment, and to illustrate some of the relationships that we will examine in data from human participants.

Consider an agent endowed with beliefs ($s^t$) about its skill in performing a task that evolve over time $t$, together with the ability to make attributions ($a^t$) for the outcomes ($o^t$) it experiences. Further suppose that it aims to learn how skilled it is. The agent repeatedly performs the task, gaining probabilistic binary feedback about its outcome on every trial ($o^t \in \{0, 1\}$). We further assume the agent uses its belief about its skill to attribute each outcome internally ($a^t = 1$) or externally ($a^t = 0$), and that it uses information from the outcome and its attribution to update its belief about its skill. This expresses the reciprocity at the heart of the ASRC.

**Generation of attributions.**   We assume that the more skilled the agent believes itself to be, the more likely it is to attribute wins internally and the less likely it is to attribute losses internally. We formalise this by letting the likelihood of internal attributions be controlled by a sigmoid function of skill for wins, and an inverse sigmoid functions for losses. Biases for internal vs external attributions are controlled by the sigmoid indifference points $x_{0w}$, $x_{0l}$, while sensitivity to skill is controlled by slope parameters $\beta_w$, $\beta_l$, for wins and losses respectively:

$$p(a^t = 1|o^t, s^t) \quad = \begin{cases} \sigma(\beta_w * (s^t - x_{0w})) & \text{if } o^t = 1 \\ 1 - \sigma(\beta_l * (s^t - x_{0l})) & \text{if } o^t = 0 \end{cases} \quad \text{where}$$

$$\sigma(x) \quad = \frac{1}{1 + e^{-x}} \quad \text{is the sigmoid function.}$$

(1)

**Skill belief updates.**   Conversely, the agent uses attributions to inform the updating of its belief about skill. On every trial, the agent computes the "prediction error", the difference between its expectation of winning given its current belief about skill and the actual outcome experienced, and corrects its internal skill estimate so as to reduce this error. The magnitude of the correction is determined by a learning rate, which we assume depends on the outcome being used for updating and on its attribution, such that both the outcome valence and the attribution modulate learning:

$$s^{t+1} \quad = s^t + \alpha_{a^t o^t} \delta_s, \quad \text{where}$$

$$\delta_s \quad = o^t - \sigma(s^t)$$

(2)

and $\alpha_{a^t o^t}$ is one of four parameters, corresponding to the previous outcome x attribution combination. Note that this simple model for skill updates implicitly captures the assumption that high skill is likely to be associated with wins, and low skill with losses, but it does not explicitly encode participant's assumptions about environment controllability, or about the relationship between their ability and outcomes, as our simulations are not aimed at investigating these assumptions.

This simplified setting includes the potential for reciprocal effects between beliefs about skill and attributions at the trial-by-trial level, and illustrates aspects of the models we will use to fit the real data, as discussed below. To illustrate the effects these interactions can have on dynamics, we performed simulations of the agent described above, varying parameters and outcomes experienced.

Fig 1A shows the skill belief evolution of 100 simulation runs of an agent with the same parameters($x_{0w} = 0$, $x_{0l} = 0$, $\beta_w = 2$, $\beta_l = 2$, $\alpha_{11} = 0.1$, $\alpha_{01} = 0.05$, $\alpha_{10} = 0.1$, $\alpha_{00} = 0.05$) and the

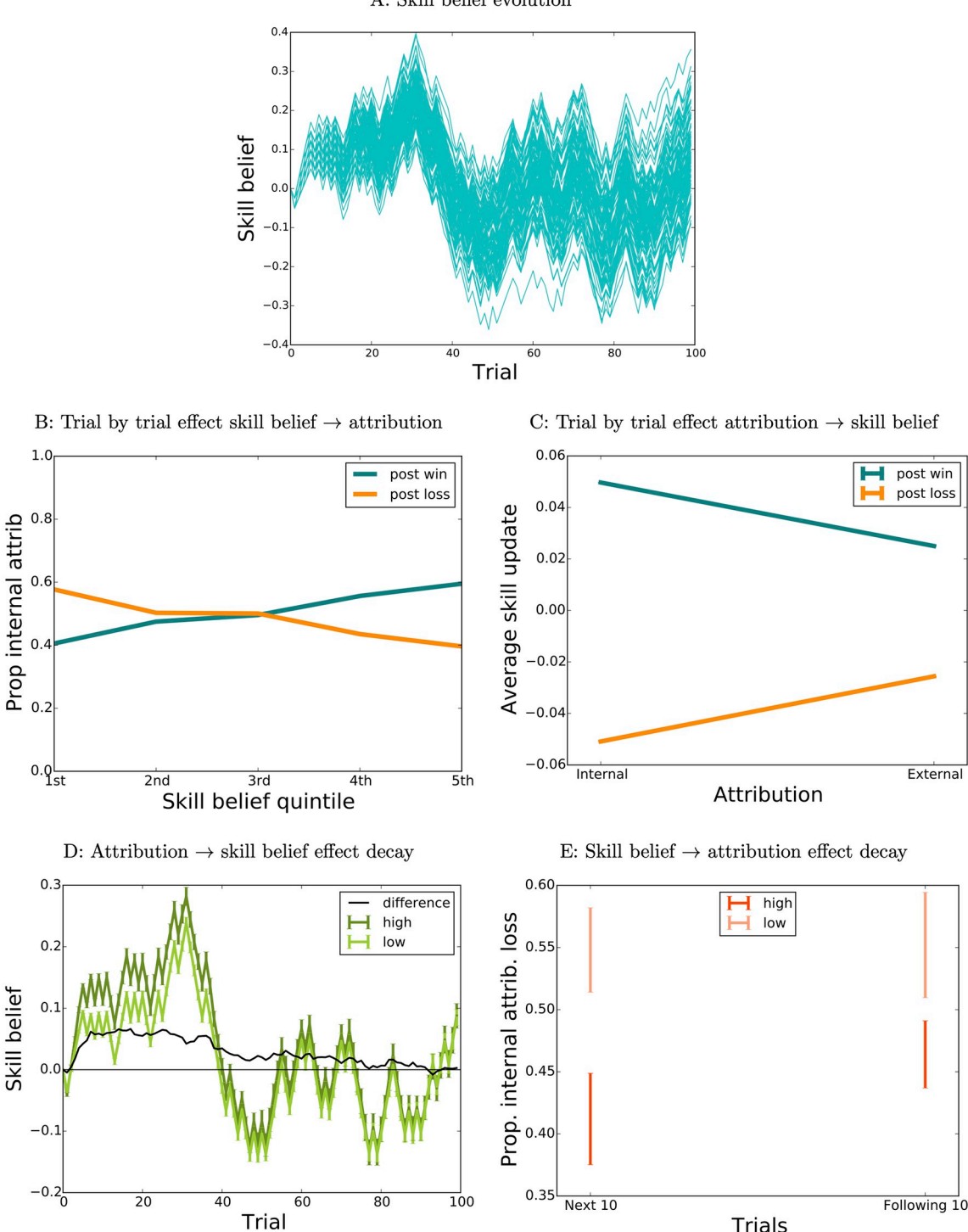

**Fig 1. Reciprocal influences between skill belief and attributions, simulation 1.** A: Skill belief evolution, 100 runs of agent with parameters $x_{0w} = 0$, $x_{0l} = 0$, $\beta_w = 2$, $\beta_l = 2$, $\alpha_{11} = 0.1$, $\alpha_{01} = 0.05$, $\alpha_{10} = 0.1$, $\alpha_{00} = 0.05$. B: Effect of skill belief on attribution at the individual trial level, across all runs plotted in A. C: Effect of attribution on skill belief update at the individual trial level, across all runs plotted in A. D: Time decay of attribution effect. E: Time decay of skill belief effect.

same initial skill belief ($s^0 = 0$), experiencing the same fixed sequence of outcomes (drawn randomly with equal probability between wins and losses on every trial), such that random variations are only introduced via the probabilistic sampling of attribution in Eq 1. The resulting variability evidently increases over trials.

The data from our human participants include just the time series of skill estimates, trial outcomes, and attributions. We might hope to unearth evidence about the underlying relationships by examining their interactions. The trial-level effect of skill beliefs on attribution is duly illustrated in Fig 1B, which shows the proportion of internal attributions for wins and losses as a function of current skill belief quintile across all trials for all the agents shown in Fig 1A. The bias to attribute wins to the self (i.e., internally) given the belief that the skill is higher, and vice-versa for losses, is apparent. Conversely, Fig 1C illustrates the trial-level effect of attribution on skill belief as the average skill belief update for each outcome x attribution combination, across the same runs. As is true of the causal process, internal attributions have a greater effect on skill belief than external ones.

Note that variability in skill beliefs is maintained and amplified despite the fact that, at least for these parameter values, differences produced by variability in early attributions decay in time, as do differences in attribution tendencies between runs with high vs low initial skill beliefs. Fig 1D shows the mean ± s.e.m of skill belief evolution for the simulation runs plotted in Fig 1A, separated (median split) according to the proportion of internal attributions for *the first 5 wins*. The effect decays steadily after trial 40, and is gone by trial 90; thus the divergence between runs is not produced by sensitivity to small random differences in initial attributions. Nor is it a result of high sensitivity to differences in skill beliefs: the effect of skill belief on subsequent attributions also decays in time, as illustrated in Fig 1E. Here simulation runs have been separated according to the skill belief value at trial 50, at which point there are already substantial differences between runs with high (dark red-top 25% of the runs) and low skill belief (light red—bottom 25% of the runs). To illustrate the decaying effect of skill belief differences on attributions we computed, for these two sets of runs, the proportion of internal attributions for losses at two later time points (left: proportion of internal attributions for the first batch of 10 losses after trial 50, right: proportion of internal attributions for the second batch of 10 losses after trial 50)—plotted in Fig 1E. Thus the sustained and even increasing divergence between the different runs is not produced in this case merely by sensitivity to initial conditions, or by a positive feedback loop. Nor is it due to randomness in attributions alone: divergence in skill beliefs is reduced in the absence of a reciprocal effect of skill belief on attributions: this is illustrated in Fig 2A and 2B, which shows (yellow) the evolution of skill beliefs for 100 agents in which attribution is matched for average propensity with the full agents (cyan), but is independent of skill beliefs (all other parameters being equal). Instead, it is the sustained mutual interactions between attributions and skill beliefs that gradually amplify the divergence between runs introduced by randomness in attributions.

For other parameter settings, reciprocal dependencies between skill beliefs and attributions can produce positive feedback loops that dramatically amplify small differences between simulation runs. This is illustrated in Fig 3, which shows 300 simulation runs of an agent starting from the same initial skill belief ($s^0 = 0$) and experiencing the same fixed sequence of outcomes as the runs in Fig 1, but with a different set of parameters($x_{0w} = -0.03$, $x_{0l} = 0$, $\beta_w = 100$, $\beta_l = 1$, $\alpha_{11} = 0.15$, $\alpha_{01} = 0$, $\alpha_{10} = 0.15$, $\alpha_{00} = 0$). Due to a combination of high sensitivity to skill belief in attribution-making ($\beta_w$, Fig 3B) and large differences between learning from internally and externally attributed outcomes ($\alpha_{11}$ vs $\alpha_{01}$, $\alpha_{10}$ vs $\alpha_{00}$, Fig 3C) differences produced by variability in early attributions are amplified, rather than dampened, in time: Fig 3D shows the skill belief evolution for the simulation runs in Fig 3A, separated (median split) according to the proportion of internal attributions for *the first 5 wins*. The effect of skill belief on attribution is

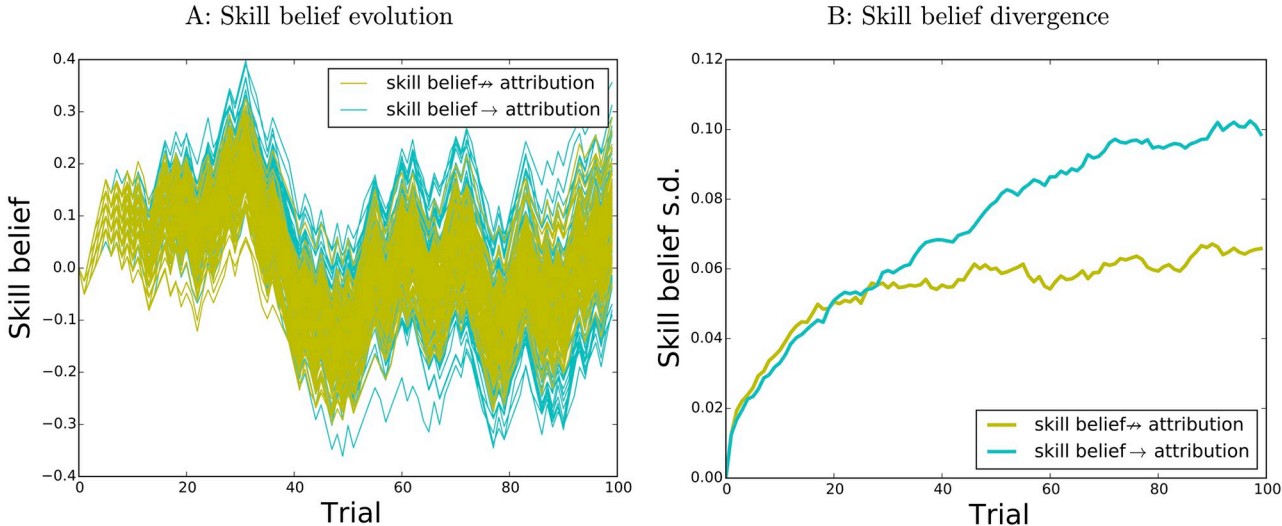

**Fig 2. Removing reciprocal connections in simulation 1.** A: Skill belief evolution, runs with (cyan, same as in Fig 1A) and without (yellow) skill belief effect on attribution. B: S. d. of runs in A.

also propagated in time, rather than decaying as in the previous example: Fig 3E reproduces Fig 1E, showing the proportion of internal attributions for two successive batches of 10 losses after trial 50, for the runs with the top 25% (dark red) and bottom 25% (light red) values of skill belief at trial 50.

Trial-level reciprocal effects also enable settings of the parameters which endow agents with "latent" vulnerability to negative outcomes—vulnerabilities that are not apparent in relatively benign conditions, when agents experience balanced positive and negative outcomes, but become apparent when they hit a streak of negative outcomes [1, 2, 28]. This is illustrated in Fig 4, which shows 100 runs of "control" agents ($x_{0w} = 0$, $x_{0l} = 0$, $\beta_w = 2$, $\beta_l = 2$, $\alpha_{11} = 0.1$, $\alpha_{01} = 0.05$, $\alpha_{10} = 0.1$, $\alpha_{00} = 0.05$) and 100 runs of "vulnerable" agents ($\beta_w = 0.5$, $\beta_l = 5$, $\alpha_{11} = 0.12$, $\alpha_{10} = 0.15$, all other parameters equal to "control") starting from the same initial skill belief value ($s^0 = 0$) and experiencing the same fixed sequence of outcomes as in Fig 1, with an additional "negative streak" of 25 losses at the end. In this particular case, compared to "control" agents, vulnerable agents are less sensitive to skill belief when making attributions for wins ($\beta_w$) and more sensitive to skill belief when making attributions for losses ($\beta_l$) (Fig 4B), as well as more sensitive to attribution when updating their skill beliefs ($\alpha_{11}$, $\alpha_{10}$, Fig 4C). This leads to vulnerable agents' beliefs growing increasingly negative with respect to their healthy counterparts during the negative streak period. This cartoon model of depression could be expanded to include agents' willingness to engage in tasks which could yield additional (positive and negative) information about their abilities [3]. If this additional component was related to the agents' current beliefs—a plausibly realistic scenario- it could further amplify differences between "healthy" and "vulnerable" agents, by impairing the latter's ability to recover from negative events even more: agents with low skill beliefs would have reduced motivation to engage in potentially rewarding activities (activities yielding positive information about the self), and could therefore get stuck in low belief states.

These simulations illustrate some of the complexity in belief dynamics which can be produced by reciprocal connections between belief updating and outcome attributions at the individual trial level. Disentangling the contributions of various parameters and or of the history of previous experience in such systems is challenging, particularly in real data, where

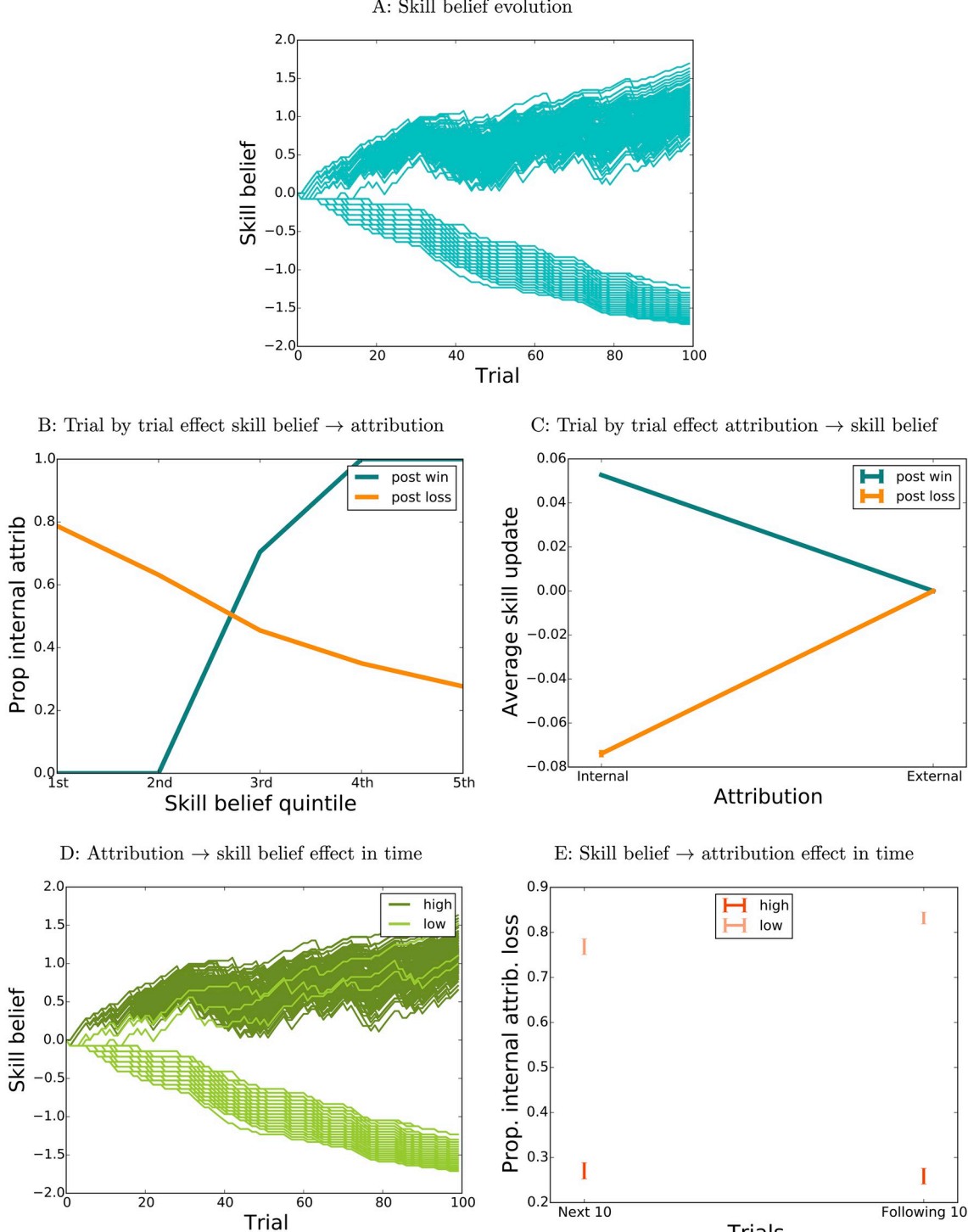

**Fig 3. Reciprocal influences between skill belief and attributions, simulation 2.** A: Skill belief evolution, 300 agents with parameters $x_{0w} = -0.03$, $x_{0l} = 0$, $\beta_w = 100$, $\beta_l = 1$, $\alpha_{11} = 0.15$, $\alpha_{01} = 0$, $\alpha_{10} = 0.15$, $\alpha_{00} = 0$. B: Effect of skill belief on attribution at the individual trial level, across all runs plotted in A. C: Effect of attribution on skill belief update at the individual trial level, across all runs plotted in A. D: Amplification of attribution effect. E: Time evolution of skill belief effect.

A: Skill belief evolution

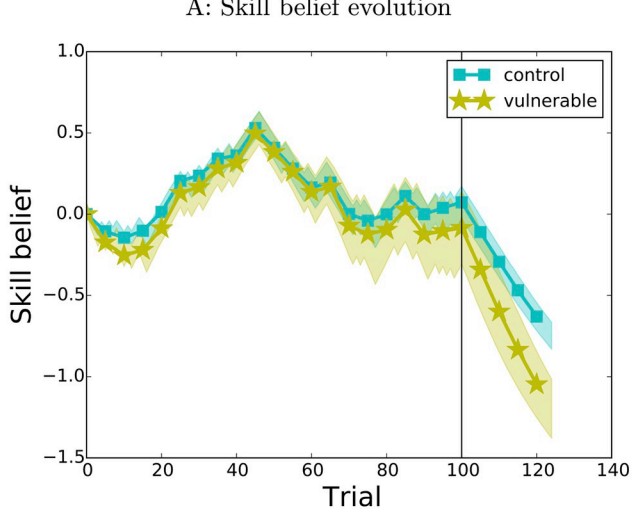

B: Trial by trial effect skill belief → attribution

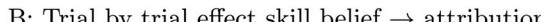

C: Trial by trial effect attribution → skill belief

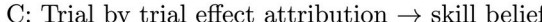

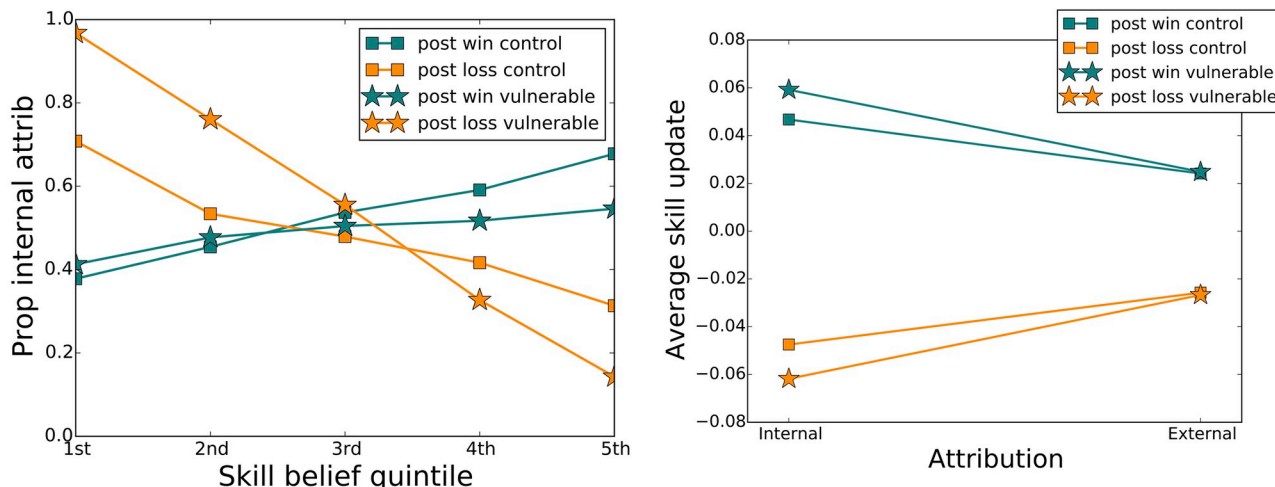

**Fig 4. Latent vulnerability, simulation 3.** A: Skill belief evolution for control (cyan, squares, $x_{0w} = 0$, $x_{0l} = 0$, $\beta_w = 2$, $\beta_l = 2$, $\alpha_{11} = 0.1$, $\alpha_{01} = 0.05$, $\alpha_{10} = 0.1$, $\alpha_{00} = 0.05$) and vulnerable (yellow, stars, $\beta_w = 0.5$, $\beta_l = 5$, $\alpha_{11} = 0.12$, $\alpha_{10} = 0.15$, all other parameters equal to control) agents. B: Effect of skill belief on attribution at the individual trial level. C: Effect of attribution on skill belief update at the individual trial level.

functional relationships between variables, as well as parameter values, need to be inferred, and history is only partially known. However such simulations can be used to provide a roadmap for dissecting real life systems and placing their study on a firm footing. In our case, the first step is establishing whether reciprocal effects between attributions and beliefs at the individual trial level can at all be detected in real life data. This is the focus of the rest of this paper.

## Experimental design

In order to investigate whether reciprocal trial-level effects between attributions and beliefs can be detected in human participants' behaviour as postulated by the ASRC, we exposed participants in our experiment to a situation similar to the one encountered by the simulated agents presented above: they had to repeatedly perform a task, evaluate their outcomes and learn about their skill.

The task we used is a remunerated game of skill inspired by the "Penguin Pursuit" game on the Lumosity "brain training" platform https://www.lumosity.com/en/. Participants are presented, on every trial, with a maze (Fig 5A) through which they need to move a token red square from the starting position to the finish position, marked by a trophy; they do so by using the arrow keys to control the token. If the token reaches the goal in the limited time allocated, a win is indicated by the appearance of a smiley face; otherwise a frowning one appears to indicate loss. During each trial, repeatedly and unpredictably, the maze rotates and the correspondence between arrow keys and the direction of movement on the screen changes, according to the following rules: the maze is equipped with a "North" direction, marked on the screen by a compass needle; the arrow keys always move the token toward the corresponding cardinal directions of the maze (Up towards "North", down towards "South", right towards "East", left towards "West"). Initially, the maze's "North" corresponds to the top of the screen; however when the maze rotates, the compass needle rotates with it, such that "North" can point to any of the four directions on the screen, and accordingly pressing the up key no longer moves the token up on the screen, but towards the maze's "North", wherever that is during each rotation. Participants therefore have to learn to adapt quickly to the change in the correspondence between key presses and resulting movements on the screen.

Trial difficulty is determined by the size of the maze, the frequency of rotations per trial and the time available. These parameters are adapted to maintain performance at roughly 50% (Fig 5B), using a double staircase procedure in order to prevent participants from gaming this adaptation. See S1 Appendix for a detailed description.

Every two trials, immediately after seeing the trial outcome, participants are asked to provide a causal attribution for the outcome and then to estimate their current skill in playing the game. Our choice of the ordering of these questions following every trial leads to a time series of alternating attribution and skill estimates responses which, in connection with our analyses, reduces potential influences of demand characteristic biases on any measured effect (see Discussion in **Reflections on the task** below). Participants are informed that a total time of 20s is available to answer each question, and that reward for the following trial would be withheld if they fail to provide an answer in this time limit. Attributions are elicited with multiple-choice questions with "ability", "maze", "rotations", "luck" as the options (see Methods section for a detailed description). These options reflect the internal vs external aspect of attribution, with the latter being separated into two different quantifiable parameters of the task (maze complexity vs rotations), and the option of blaming or crediting luck. Skill estimates are reported with a slider on a continuous scale from "very bad" to "very good" (Fig 5A).

The experiment involves two conditions: first participants play the game themselves and answer questions related to their performance; we refer to this as the self condition. Participants are then told that they will watch trials recorded from another participant having previously participated in the experiment and they are asked to make attributions, evaluate skill and bet on this "other"'s performance; this is the other condition. In fact, each participant is shown their own recorded trials from the self condition, to control for the precise sequence of difficulties and outcomes encountered. In order to reduce the likelihood of participants recognising their own performance, mazes are left-right mirrored when played back. Due to the length of the task, each condition was split into two sessions, performed on two successive days; there was a 7 day delay between conditions.

Note that as the ASRC only refers to the self, we did not have explicit hypotheses regarding the "other" condition. There is, however, ample research on distinctions between self and other within the attribution literature, notably the actor-observer effect [37] (see [38], for a review). Our aim in introducing this condition was to specifically explore relationships

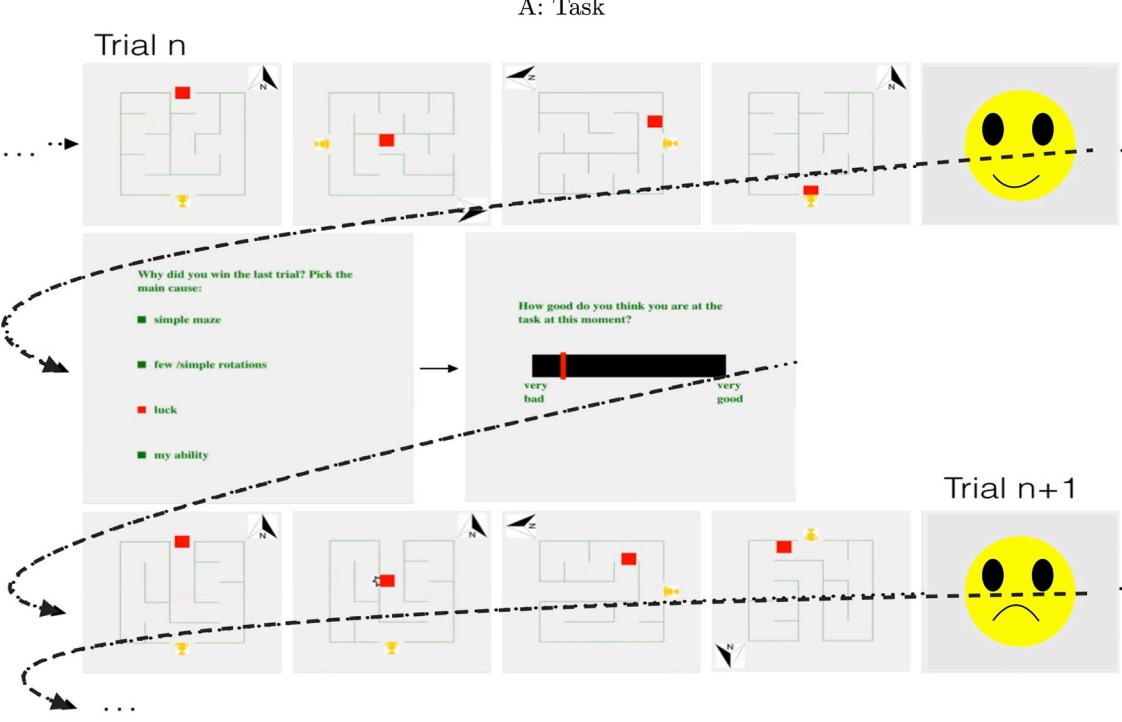

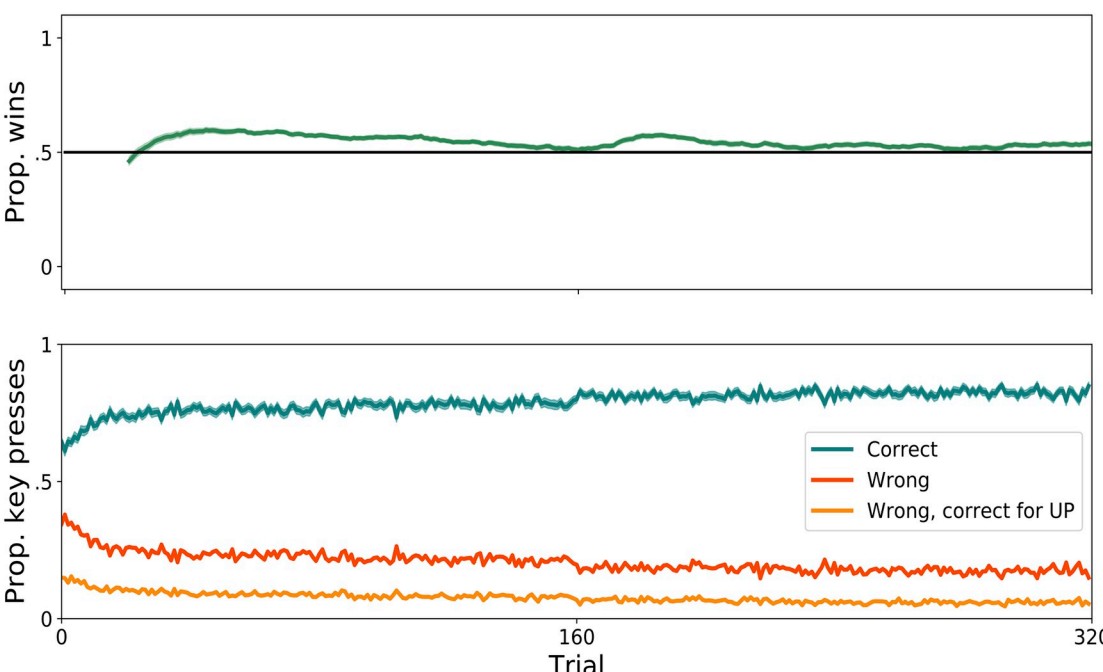

**Fig 5. Task and performance overview.** A: Task structure and example trials: four frames sampled for illustration purposes from two trials are displayed in order on the top and bottom rows. After every two trials, participants are asked to attribute the latest outcome to one of 4 given causes, and then to report how good they believe themselves to be at the task. Dotted arrows indicate the flow of time. B: Evolution of performance across trials, mean ± s.e.m. across participants; top: running average of the proportion of wins, sliding window 20 trials, bottom: per trial proportion of correct key presses, wrong key presses and wrong, but correct for the normal UP orientation.

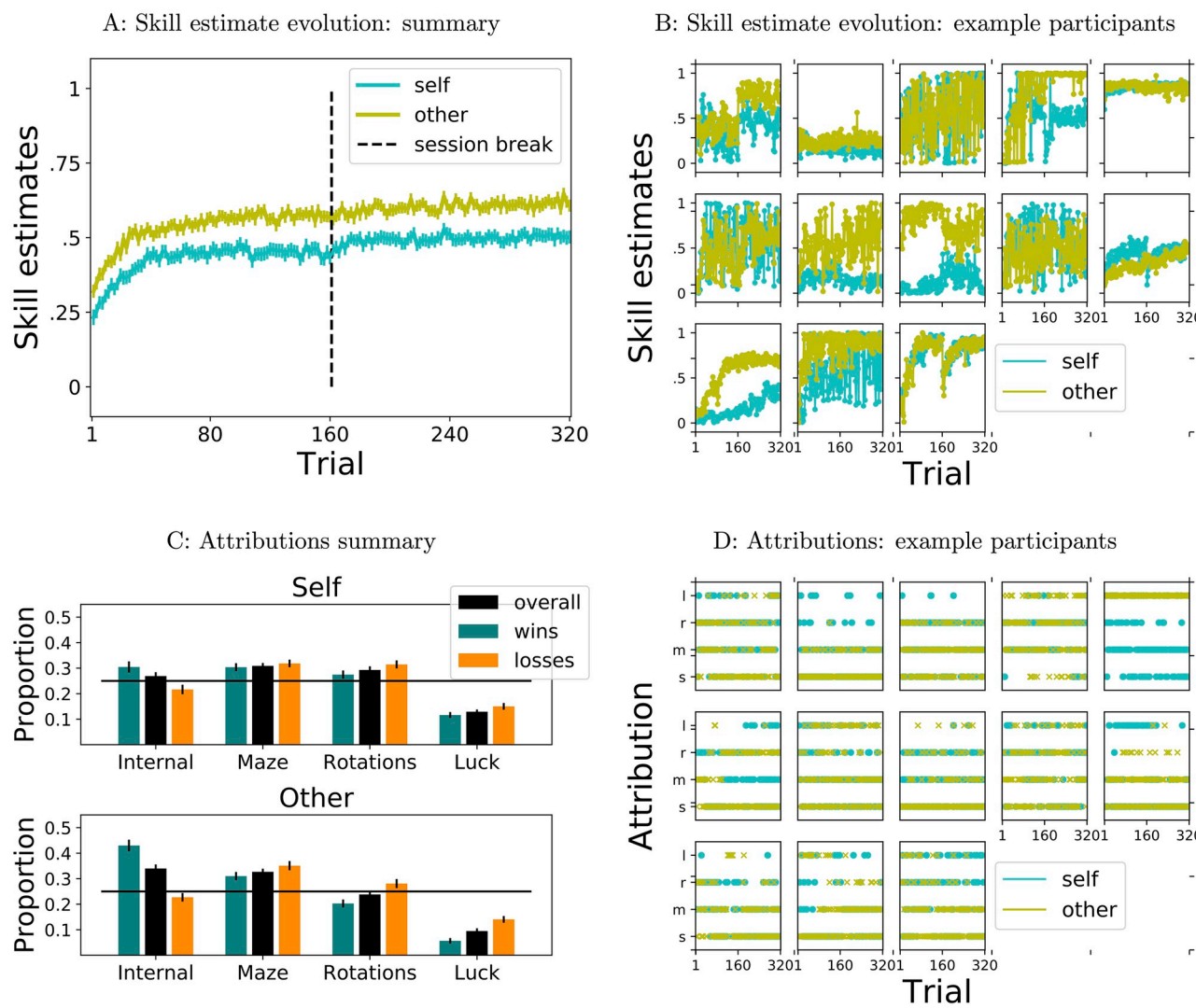

**Fig 6. Skill estimates and attributions overview.** A: Evolution of skill estimates across trials, mean ± s. e. m. across participants. B: Evolution of skill estimates for individual participants, chosen to illustrate variability. C: Attribution proportions, mean ± s.e.m across participants, overall and conditioned on outcomes. D: Time series of attributions for individual participants, chosen to illustrate variability, not the same as in B.

between causal attributions and belief updating when another is involved, and compare the "self" and "other" conditions.

Three questionnaires were administered immediately prior to the first session: the Attributional Style questionnaire (ASQ) [9], the Levenson Locus of Control questionnaire [39] and the Rosenberg Self esteem scale [40].

Fig 6 shows a summary of participants' skill estimates and attributions (A, C), as well as full time series of responses for a selection of individual participants (B, D), illustrating individual differences.

## Analyses of skill estimates

We performed both model-agnostic and model-dependent analyses in order to test whether participants' attributions had detectable effects on their skill estimates; we hypothesised that

outcomes attributed internally would have a larger impact on beliefs than outcomes attributed externally.

**Model agnostic analyses.** As expected, we found an effect of outcome on skill estimate updates, with significantly larger updates after wins than after losses for both self (paired $t = 14.91$, two-sided $p < 1/5000$, $d = 2.65$, see Methods section for details) and other (paired $t = 15.24$, two-sided $p < 1/5000$, $d = 3.1$), Fig 7. We therefore tested for an effect of attribution (internal vs external) on skill estimate updates conditioned on outcome and found significant effects of attributions on both wins (self: paired $t = 4.14$, two-sided $p < 1/5000$, $d = 0.52$; other: paired $t = 7.09$, two-sided $p < 1/5000$, $d = 1.01$) and losses (self: paired $t = −3.47$, two-sided $p = 0.0008$, $d = −0.43$; other: paired $t = −4.06$, two-sided $p < 1/5000$, $d = −0.62$) in the expected directions: participants' reported skill estimates increased more after wins attributed internally

A: Outcome, attribution → skill estimate updates

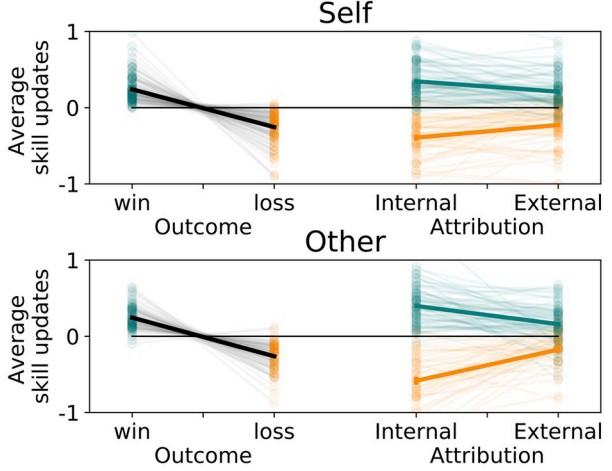

B: Winning skill model parameters self

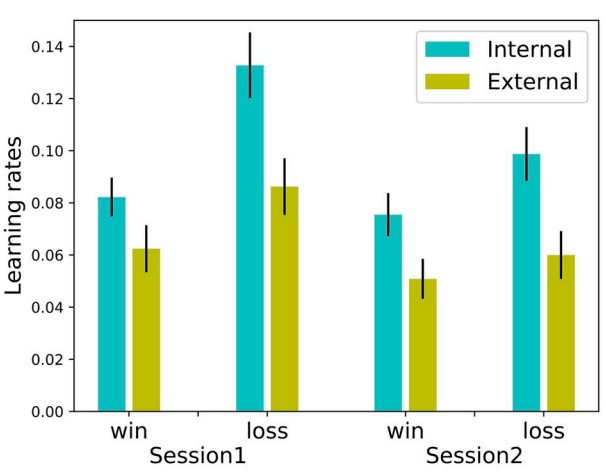

C: Skill estimates model comparison

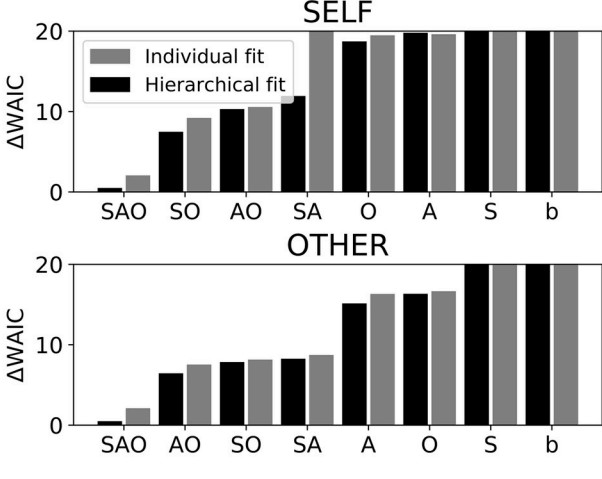

D: Winning skill model parameters other

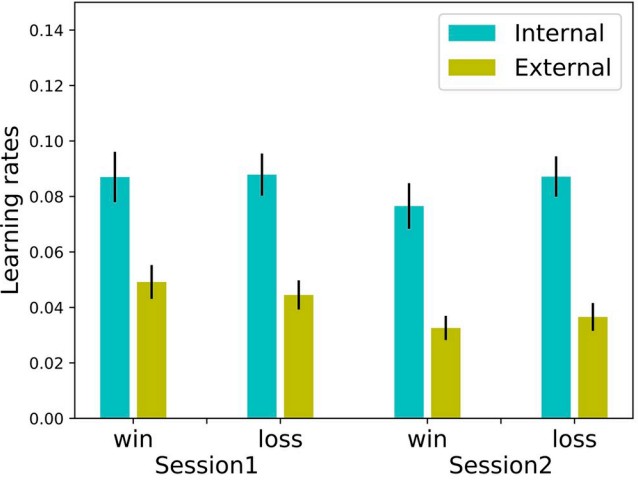

**Fig 7. Effect of outcome and attribution on skill estimate updates.** A: Model agnostic analyses: faded lines and dots represent individual participants, bold lines represent mean± s.e.m across participants. B:, D: Learning rates, winning model for skill. B: self, D: other. Cyan: learning rates for internal attribution, yellow: learning rates for external attributions. C: Skill estimates model comparison; baseline Rescorla-Wagner model (b), and augmentations with learning rates varying based on session (S), attribution (A), outcome (O) and combinations of these factors. Top: self, bottom: other. Difference in WAIC scores from each model to the preferred one. Smaller WAIC scores indicate better models.

than after wins attributed externally, and decreased more after losses attributed internally than after losses attributed externally (Fig 7A).

**Model comparison.**   We next turned to trial-by-trial models of participant's skill estimates. We compared variants of Rescorla-Wagner models, built on the assumption that participants maintain an internal estimate of skill, which they adjust as a result of the outcomes they experience (in the self condition)/they observe (in the other condition). Thus on every trial the prediction error—the difference between the outcome and the current estimate of skill—is used to update the underlying skill estimate. The factor weighing the prediction error's contribution—the learning rate—is what differs between models (see complete model description in Methods section). Due to our main interest in the effect of attributions on skill estimates, one change to the baseline model was to allow the learning rate to be different for different attributions (model A). Because participants' responses were not constrained, attribution and outcome could not be entirely orthogonalised in the data, and participants showed a preference for internal attributions for wins and external attributions for losses (see **Analyses of attributions**); therefore to control for any effect of outcome disguising as an effect of attribution in model A, we also allowed learning rates to be different for wins vs losses (model O). Finally, because each condition was divided between two successive days (see **Experimental design**), we introduced an additional variation to the baseline model by allowing learning rates to differ between the two sessions (S). We therefore compared a family of models in which learning rates were allowed to vary orthogonally along three directions- outcome, attribution, session-leading to a total of 8 models. See Methods section for full details on the model fitting and model comparison procedures.

For both self and other, model comparison favoured a hierarchical version of the full model (self: WAIC = -92.15, ΔWAIC = -6.99 to the next best model, other: WAIC = -104.86, ΔWAIC = -5.95 to the next best model, Fig 7C), indicating that allowing learning rates to differ for internal vs external attributions improved the model evidence, despite a higher penalty for the increased number of parameters. The best model was well able to fit the data (self: $r^2 = 0.44 \pm 0.24$ across participants, worst $r^2 = 0.01$, best $r^2 = 0.93$; other: $r^2 = 0.42 \pm 0.24$, worst $r^2 = 0.01$, best $r^2 = 0.95$; see S3 Fig for example best and worst participant fits).

**Model parameters.**   Results of model-agnostic analyses indicated that outcomes attributed internally had a stronger impact on participants' subsequent skill estimates than outcomes attributed externally. Analyses of parameters from the best skill model confirmed this result (Fig 7B and 7D): for both self and other we found higher learning rates for internal than for external attributions (self: paired $t = 8.63$ one-sided $p < 1/5000$, other: paired $t = 12.08$, one-sided $p < 1/5000$, p-values computed from permutation tests). In fact, the difference between learning rates for internal and external attributions was significant for each outcome x session combination (see S1 Table for full results). This result was also confirmed by analyses of fits with shuffled attribution responses (see S4 Fig for details and results).

In addition, learning rates were lower for wins than for losses for self (paired $t = -6.34$, two-sided $p < 1/5000$ from permutation test), but not for other (paired $t = 0.82$, two-sided $p = 0.41$); this effect of outcome in the self condition was stronger for internal than for external attributions (paired $t = -2.65$, two-sided $p = 0.008$).

Thus both model-agnostic and model dependent analyses identified significant effects of attribution on skill estimates, outcomes attributed internally having a stronger effect on beliefs than those attributed externally. Note that these effects are detected on *updates* in skill estimates, rather than on skill estimates reported immediately following attribution, which therefore reduce potential contributions of demand characteristic effects. We cannot, however, entirely rule out such contributions. In addition, model-based analyses indicated stronger

learning from losses than from wins in the self condition, particularly in the case of internally attributed outcomes.

## Analyses of attributions

We next performed both model-agnostic and model-dependent analyses of attributions. Our main question was whether participants' skill estimates had detectable effects on their attributions; our hypothesis was that participants would make more internal attributions for wins and less internal attributions for losses when believing they possess high rather than low skill. We also investigated the effects of objective task features and performance, which we expected to be reflected in attributions to the relevant response options provided; specifically, we expected to find effects of path length on attributions to maze, and effects of the frequency of rotations on attributions to rotations. All statistical tests were performed using permutations to check for significance, as detailed in the methods section.

**Model-agnostic analyses.**   Consistent with previous results [21, 30] we found that overall participants made more internal attributions for wins than for losses in both conditions (self: paired $t = 3.1$, two-sided $p = 0.0036$, $d = 0.54$; other: paired $t = 6.69$, two-sided $p < 10^{-5}$, $d = 1.07$, p-value computed from permutation tests, see Methods section for details), with a significantly larger effect for other than for self (paired $t = 3.77$, two-sided $p = 0.0004$, $d = 0.42$). Analyses of the effects of objective task and performance measures on participants' attributions to the relevant attribution options revealed that participants were sensitive to the task manipulations, as well as to their own performance. Participants made more attributions to the maze for losses and fewer for wins as the length of the correct path through the maze increased (significant interactions of path length with outcomes on attributions to maze: self: $F = 190.03$, $p < 1/5000$, other: $F = 127.12$, $p < 1/5000$, Fig 8A for self and S5 Fig for other; F statistics computed as for two-way repeated measures ANOVA [41], with p-values estimated from permutation tests, see Methods section). Similarly, the more the maze was displayed in an unusual orientation during the trial, the more participants blamed rotations for losses and the less they credited rotations for wins (significant interactions of prop non up orientations with outcomes on attributions to rotations: self: $F = 37.71$, $p < 1/5000$, other: $F = 22.02$, $p < 1/5000$, Fig 8A for self and S5 Fig for other). Participants' attributions were also sensitive to objective performance: participants increased their assignment of losses to bad luck and decreased crediting wins to good luck with increasing accuracy in their key presses (significant interactions of prop correct key presses with outcomes on attributions to luck: self: $F = 53.23$, $p < 1/5000$, other: $F = 9.12$, $p = 0.0006$, Fig 8A for self and S5 Fig for other). These results confirm the effectiveness of task manipulations, and show that participants understood the attributions options provided, using them reasonably.

Analyses of the effect of skill estimates on internal attributions confirmed our hypotheses, identifying significant interactions between outcome and previous skill estimate (self: $F = 33.92$, $p < 1/5000$, other: $F = 29.51$, $p < 1/5000$, Fig 8A for self and S5 Fig for other), with more internal attributions for wins and less internal attributions for losses with increasing skill estimates.

**Model comparison.**   In model-agnostic analyses we considered the effect of several variables on attributions *separately*. We next turned to trial-by-trial models of participants' attributions in order to investigate the effects we observed in model-agnostic analyses while controlling for the other factors. Here as well, we were mainly interested in the effect of reported skill estimates on attributions.

We used linear classification models, in which the contributions of the features of interest are first linearly combined into a score for each response option $s_o = \boldsymbol{w}_o \cdot \boldsymbol{f}$, with option-specific

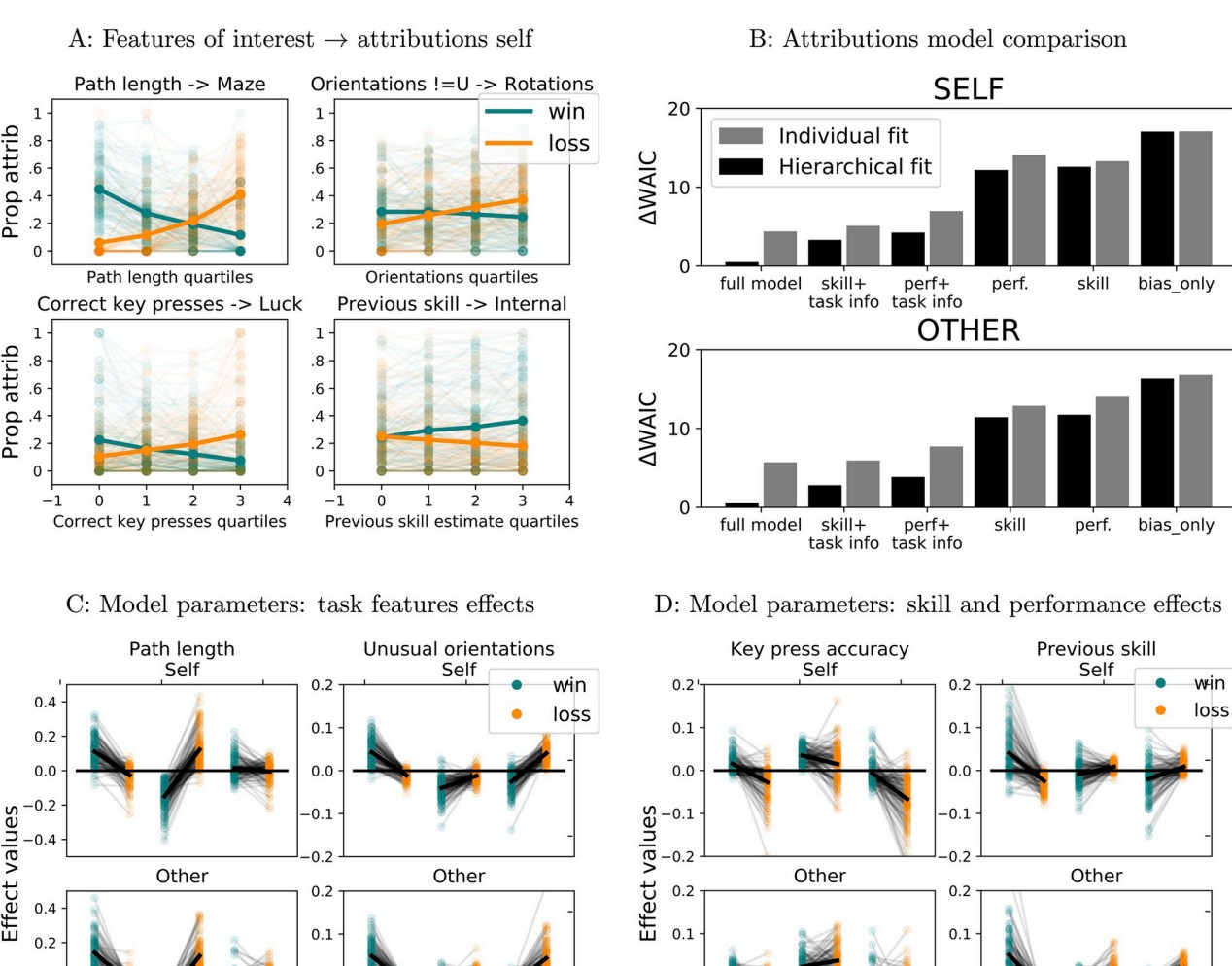

**Fig 8. Attribution analyses.** A: Features of interest (skill estimates, task features, objective performance) and attributions summary self: faded lines represents individual participants, bold lines represent mean ± s.e.m across participants. Orange: losses, teal: wins. B: Attribution model comparison. Top: self, bottom: other. Difference in WAIC scores from each model to the preferred one. Smaller WAIC scores indicate better models. C: Winning attribution model parameters. Effects of path length (left) and proportion of unusual orientations (right). Top: self, bottom: other. D: Effects of key press accuracy (left) and skill estimates (right). Top: self, bottom: other.

weights $\boldsymbol{w}_o$ as parameters, and then the resulting scores are passed through a softmax function to obtain response probabilities for each option $p(o) = \frac{\exp(s_o)}{\Sigma_{o \in O} \exp(s_o)}$ (see Methods section for full model specifications). Because at least some of the features are expected to have different, potentially opposite effects on attributions for wins vs losses, as indeed it emerged from model-agnostic analyses of the data, all models were equipped with separate parameters for wins and losses. Because participants have to choose one of the 4 available response options, models were equipped with independent parameters for *three* of the options, and scores for the 4 options were constrained to sum to 0; thus parameters and preference for the "Luck" option were entirely derived from parameters and preferences for the other options, and are not analysed nor reported separately below. The models we compared varied in that they

included different potential features: no features—baseline propensity for each attribution option; models including bias and one of the following sets of features: skill estimates, performance features, performance features and task features, skill estimates and task features; and the full model: skill estimates, performance features, task features. See Methods section for a detailed account of the models, model fitting and model comparison procedures.

For both self and other, model comparison favoured the full model in its hierarchical version (self: WAIC = 155.21, ΔWAIC = -2.81 to the next best model, other: WAIC = 145.38, ΔWAIC = -2.29 to the next best model, see Fig 8B), indicating that including skill estimates as a feature over and above objective performance and task features improved the model evidence, despite a higher penalty for the increased number of parameters. The winning model was able to predict participants' responses above chance (chance level = 0.25, median probability of true answer self: 0.48 ± 0.14, ranging from: 0.27 to 0.93; other 0.53 ± 1.64, ranging from 0.28 to 0.99).

**Model parameters.** We then performed analyses of model parameters from the winning attribution model. These fall into two categories: baseline preferences for different attribution options, as captured by bias parameters, and the effects of various features, as captured by the corresponding feature weights in the model (in both cases, direct comparisons between raw parameters are not meaningful, due to the presence of different sigmoid functions for wins and losses; all effects reported were obtained by transformations of the raw parameters to correct for this, as detailed in Methods section). These confirmed the results of model-agnostic analyses, but also revealed additional insights, as detailed below.

Consistent with model agnostic results, baseline preference for internal attributions was significantly higher for wins than for losses in both conditions (self: paired $t = 4.77$, two-sided $p < 1/5000$, $d = 0.61$, other: paired $t = 10.34$, two-sided $p < 1/5000$, $d = 1.36$), with a significantly stronger effect for other than for self (other vs self paired $t = 4.65$, two-sided $p < 1/5000$, $d = 0.58$).

Recapitulating the results of model-agnostic analyses we found that for both self and other increasing path length significantly reduced the likelihood of attributing wins to the maze and significantly increased the likelihood of attributing losses to the maze, with the opposite pattern for internal attributions (Fig 8C, results in S2 Table). Similarly, increasing proportion of unusual orientations of the maze during trial significantly reduced the likelihood of attributing wins to rotations and increased the likelihood of attributing losses to rotations, with the opposite pattern for internal attributions (Fig 8C, results in S3 Table).

The effects of skill estimates on internal attributions were also in the expected directions, increasing the likelihood of attributing wins internally (self: $m = 0.04$, two-sided $p < 1/5000$ from sign permutation test; other $m = 0.05$, two-sided $p < 1/5000$) and decreasing the likelihood of attributing losses internally (self: $m = -0.03$, two-sided $p < 1/5000$; other $m = -0.04$, two-sided $p < 1/5000$; Fig 8D). Skill estimates had qualitatively similar effects on the two external options: increasing skill estimates increased the likelihood of blaming maze and rotations for losses and decreased the likelihood of crediting them for wins (Fig 8D, results in S4 Table). This is different from the effects of momentary performance: increasing trial-level key press accuracy increased the likelihood of attributions to maze and decreased the likelihood of attributions to rotations irrespective of outcome (Fig 8D, results in S5 Table).

Thus both model-agnostic and model dependent analyses identified skill estimates as having significant effects on participants' attributions. In addition, model-dependent analyses revealed differences between skill estimates and key press accuracy in the pattern of their results on external attribution options; these are consistent with the distinction between a momentary and specific performance measure on the one hand, and a more stable and general measure of ability on the other.

### Relationships with questionnaire scores

In order to test whether our task taps into the attribution and belief dimensions typically investigated with well-established questionnaire measures we compared participants' questionnaire scores with summary statistics of their responses in our task and individual parameters from winning models.

**Questionnaires and behaviour in the task.** We first investigated relationships between questionnaire measures and aspects of behaviour in our task that seem intuitively related to them. We hypothesized that self esteem scores would be related to patterns of self assessment in both skill and attribution responses: specifically, we expected participants with higher self-esteem scores to judge themselves as more skilled, to show larger differences in skill estimate updates after wins vs after losses, and to show larger differences between the proportions of internal attributions for wins vs losses, compared to participants with lower self-esteem. We also investigated the relationship between internal locus of control and the proportion of internal attributions, and that between scores on the internality positive and internality negative subscales of the ASQ and the corresponding proportions of internal attributions in our task.

We computed correlations between the relevant behavioural variable and relevant questionnaire score for or each of these hypothesised relationships and tested significance by performing permutation tests. For self esteem (SE), we found a significant positive correlation with the average skill estimate ($r = 0.27$, two-sided $p = 0.0026$, Fig 9A), and a significant positive correlation with the difference between the proportions of internal attributions for wins and losses ($r = 0.25$, $p = 0.0074$, Fig 9B), both in the expected directions; there was no significant relationship with the difference in skill estimate updates after wins vs after losses ($r = 0.14$, $p = 0.1086$, Fig 9C). There was a positive correlation between the internal locus of control subscale and the proportion of internal attributions in the task ($r = 0.19$, $p = 0.0464$, Fig 9D). Finally, the proportion of internal attributions post losses in the task was positively correlated with the Internality for negative events subscale of the ASQ ($r = 0.24$, $p = 0.016$, Fig 9E); the relationship between the proportion of internal attributions for wins and the Internality for positive events subscale of the ASQ was not signficant ($r = 0.15$, $p = 0.1408$, Fig 9F).

**Questionnaires and model parameters.** These relationships were confirmed by analyses of the related model parameters. Thus we found a positive correlations between SE and the initial skill parameter ($r = 0.23$, $p = 0.0284$), the difference between learning rates for wins and learning rates for losses ($r = 0.22$, $p = 0.0328$; see Methods section for details of parameter computation) and the difference between learning rates for internal attributions for wins vs losses ($r = 0.27$, $p = 0.0058$) in the winning skill model; SE was also positively correlated with the difference between the baseline preference for internal attributions post wins vs losses extracted from the winning attribution model ($r = 0.26$, $p = 0.0098$). Scores on the internal subscale of the locus of control questionnaire (LCi) were positively correlated with the baseline preference for internal attributions ($r = 0.21$, $p = 0.0446$), and scores on the internality for negative events subscale of the ASQ were positively correlated with the baseline preference for internal attributions for losses ($r = 0.28$, $p = 0.0052$). The analogous relationship for positive events was not significant ($r = 0.15$, $p = 0.1276$).

## Discussion

### Summary

We developed a novel task to test for the presence of trial-level reciprocal effects in the generation of causal attributions and beliefs about self, as such high temporal resolution effects are central to attribution-self-representation cycle theory [1, 2] and its account of the generation

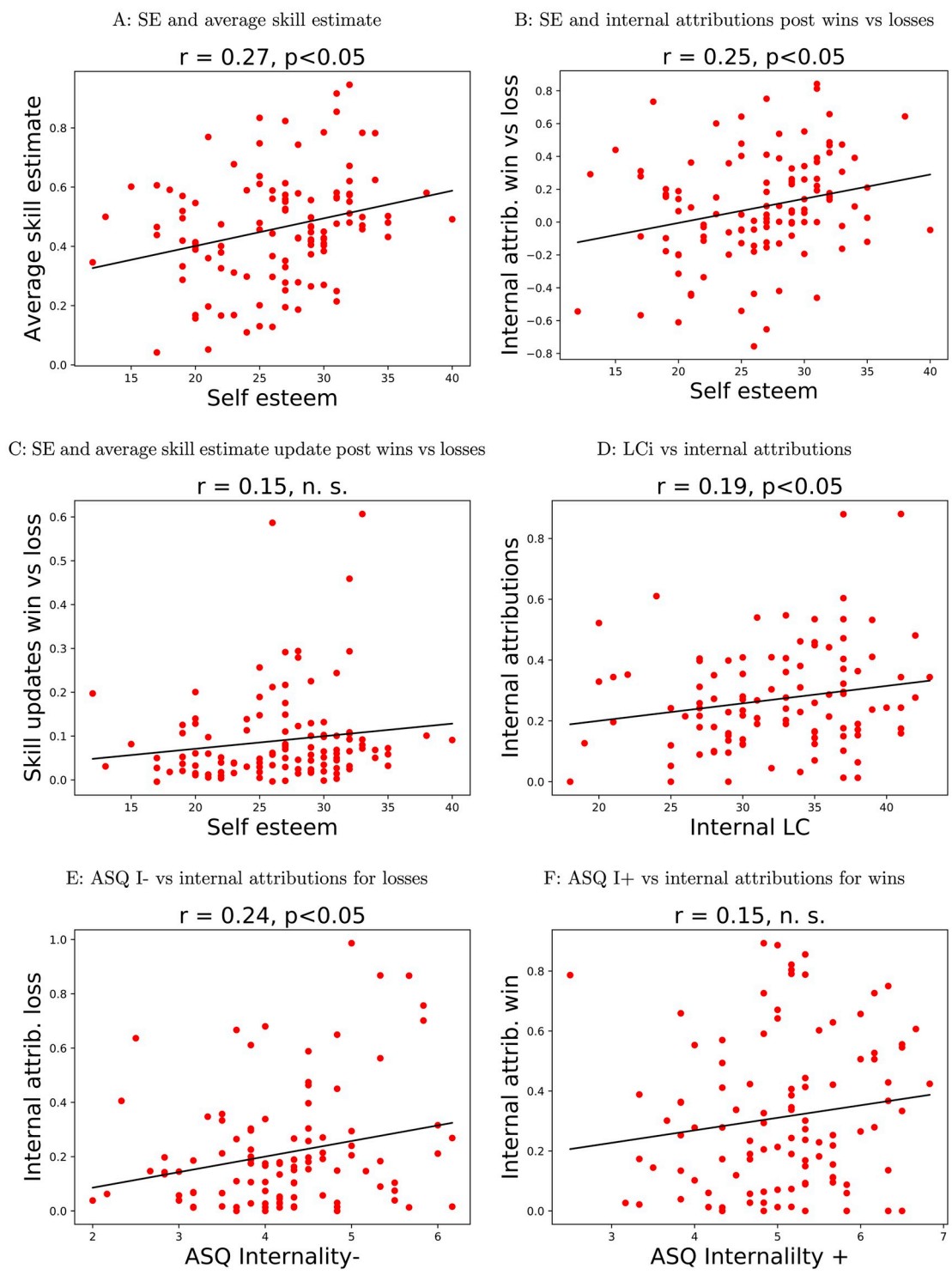

**Fig 9. Behavioural measures vs questionnaire scores.**

and maintenance of psychiatric disorders. In the context of learning a new competence, participants repeatedly experienced real outcomes and reported both causal attributions and beliefs about skill (skill estimates). In accordance with the theory, and at a trial-by-trial level, participants changed their skill estimates more after outcomes attributed internally than after ones attributed externally. Conversely, with increasing skill estimates, they took more credit for wins and less blame for losses, showing that attributions also varied with beliefs. Correlations between behaviour in the task and questionnaire-based measures of self-esteem, control, and attributional style suggest that the task indeed taps into some of the mechanisms investigated by these tools.

Model-dependent analyses confirmed these results: model comparison favoured a skill model with higher learning rates for internally attributed outcomes than externally attributed ones, and a model for attributions which included skill estimates as a feature over and above objective task and performance measures. Analyses of model parameters recapitulated model-agnostic results, and provided additional insights into the mechanisms at work, notably an effect of valence on learning rates for self, but not for other, distinctions between the effect of trial-by-trial performance measure vs estimated skill on attributions, and relationships between individual differences in parameters and questionnaire scores.

Our results provide the first evidence at a high temporal resolution of the dynamical nature of, and interactions between, attribution and beliefs about skill, and lay the foundations for computational investigations of the behavioural regimes that such reciprocal interactions can sustain.

## Reflections on the task

The ASRC postulates continuous reciprocal influences between beliefs and attributions, which are conceived as alternating time series, i. e attributions and belief updating constantly both follow and precede each other. For implementation purposes we needed to discretize the sampling of these variables, probing attributions and beliefs on a trial-by-trial basis. While we cannot completely exclude the possibility that these prompts might have influenced participants' responses, our design reduces the likelihood of such effects contaminating our results. We chose the ordering of the attribution and skill questions following every trial so as to obtain a time series of alternating attributions (A) and skill estimates (S) responses ($A^0$, $S^0$, $A^1$, $S^1 \ldots A^T$, $S^T$) which reduces influences of demand characteristic biases on the effects of interest in our analyses: the effect of attribution was measured on skill estimate *updates* (i. e. effect of $A^t$ on $S^t - S^{t-1}$), rather than on the immediately following skill estimate report; conversely, the effect of skill estimates was measured on the attribution collected after the following trial (i. e effect of $S^t$ on $A^{t+1}$). Under the alternative question ordering we would have obtained the alternative time series, $S^0$, $A^0$, $S^1$, $A^1$, $\ldots S^T$, $A^T$, and measuring the effect of skill estimates on attributions would have involved the effect of $S^t$ on $A^t$, more prone to corruption by demand characteristic effects.

The novel task that we introduce in this work differs from previous work in several important ways. Participants' own attributions for outcomes and beliefs about skill are collected at fine temporal granularity, in the absence of any manipulation targeting their content. These attributions and skill estimates refer to real outcomes, which depend on participants' performance as well as on objective task manipulations, and which are experienced in a gameified context that promotes engagement and enhances motivation. While understanding the rules and playing the game are straightforward for participants, disentangling the extent to which various task aspects and the player's own performance contributed to the outcome is not. Participants are therefore faced with a realistically challenging problem in interpreting their experience to learn about how good they are at the task.

These novel aspects of the task enabled the investigation of trial-level effects of causal attributions on beliefs and the reverse, in a relatively realistic scenario involving real outcomes, beliefs and attributions, in a large population of online participants. There were, however, a number design choices that we could not readily base on evidence from previous work, and which could be improved in the future. Most importantly, we did not have access to any previously validated objective measures of difficulty and skill; instead, in order to maintain the balance between positive and negative outcomes, as well as to maintain subjects' engagement in the task, we adapted trial difficulty to subjects' performance level with a staircase procedure, involving the various aspects of the task that were, intuitively, the most relevant; we then attempted to extract a post hoc objective measure of difficulty from the data (see S2 Appendix). The staircase procedure did achieve the practical desiderata of satisfying time constraints and providing balanced numbers of wins and losses. However, the steps of difficulty for some of the components were rather coarse, implying undesirably high trial-to-trial variations in difficulty (see S2 Appendix). Given the evident viability of the framework, it would be possible in future work to explore at a finer granularity the components of the task that determine difficulty and to calibrate them by measuring the frequency of wins and losses for each setting of these components in a large population of subjects.

In all our analyses of participants' data, "skill" always referred to participants' own skill estimates. There are, however, several important distinctions to be made regarding the notion of skill: the timescale distinction between skill and momentary performance—a skilled player might still make mistakes and hence produce poor momentary performance, that between objective skill and perceived skill and the subtler issue of actually defining objective skill in relationship with objective difficulty—a skilled player is one who can prevail in difficult circumstances, while both a skilled and a less skilled player can win easy trials. Due to the nature of the task and the challenges of defining and measuring difficulty, we did not define, measure, discuss or analyse objective skill in this paper. This also precluded us from investigating relationships between objective and perceived skill, and relationships between any such discrepancies and attributions. Using an externally validated measure of difficulty together with a precisely calibrated staircase would allow participants' real underlying skill level to be closely tracked and objectively measured, enabling investigations into relationships between the accuracy of participants' beliefs about skill and their patterns of attributing outcomes, or questionnaire-based psychological measures. A number of questions that we could not address in this work could then be asked, such as whether participants display self-serving biases in their skill estimates, and, if so, whether such biases are associated with self-serving attribution patterns or with higher self-esteem scores.

Finally, better empirical knowledge of task psychophysics could conceivably enable artificial simulation of agents with specified properties, which would remove the need to deceive participants about the other condition and allow the ordering between conditions to be balanced, as well as enable additional questions to be tackled. Introduction of multiple 'other' conditions could be used to control for the effect of watching vs playing; evaluation of close "others" could be compared with evaluation of indifferent, hostile or artificial "others"; other social aspect of the processes involved could also be investigated, by e.g. introducing competition into the task, or by providing agents with a learning environment populated by real or artificial peers.

## Self-serving and self-defeating mechanisms

Attributions in our data showed evidence for both self- and other-serving biases: in both conditions, participants made more internal attributions for wins than for losses. This pattern, in which people take more credit for positive outcomes than they take for negative ones, has

often been reported before [13–15, 27, 28] (see [21, 30] for reviews), and is thought to contribute to the maintenance of well being, promoting persistence and exploration in the face of negative feedback or failure.

Selective attendance to, or overweighting of, positive information has been found in studies investigating self-evaluation or the processing of other self-relevant information [29, 42] (see [21, 30, 31] for reviews), where presumably it reflects biases toward maintaining a positive view of oneself, as well as in a large number of RL studies [43–48] (though the opposite pattern has also been reported, see [49, 50]). Note that significant differences are likely to exist between both credit assignment and motivation mechanisms in RL vs self-evaluation tasks such as ours, and that learning rates in the two cases are of different natures: they reflect changes in preference for different choices in the former, and changes in self-related beliefs in the latter. Indeed in our data we observed heightened learning from negative feedback for self (but not for other): learning rates for skill estimates associated with losses were higher than those associated with wins, with a larger effect for internally attributed outcomes than for externally attributed ones.

There is also ample evidence in animals as well as humans of the privileged processing of negative feedback: negative, more than positive, feedback produces rapid and strong bodily responses, mobilising the organism for reaction; negative emotions produce more arousal than positive ones; negative events and information focus attention (see [51] for a review); in humans, concepts for negative actions and consequences form earlier than their positive counterparts [52], negative events are surveyed more for potential causal information [53, 54], and they elicit more spontaneous causal attributional activity than positive ones [55]. Our result replicates that of [56] who, in a learning context somewhat similar to ours, found higher learning rates for negative than for positive feedback, for self but not for other. The fact that in our data the effect was larger for internally attributed outcomes is consistent with an emphasis on negative feedback to improve performance while learning a novel task, since learning from internally attributed failures could be particularly informative for improving future performance. Heightened attention to negative feedback might be particularly relevant in complex environments requiring extended sequences of actions, as such scenarios involve multiple aspects of performance, making learning about and improving aspects of one's performance both crucial and challenging [56, 57]. Our experiment was not designed for the purpose of testing the effect of environment demands on learning rate adaptability in learning contexts and is not suitable for such a test, but our observations suggest this would be an interesting goal for future work.

The apparently contradictory pattern of biases in attribution and learning might be involved in maintaining balance between learning from negative feedback and maintaining a positive view of the self. In this scenario, the bias toward internal attributions for positive outcomes counteracts the threatening effects of heightened sensitivity to negative feedback, while this sensitivity promotes efficient learning from undesirable outcomes even in the presence of opposing motivational and emotional biases.

## Loopy dynamics

As the name clearly states, the core concept of the attribution self-representations cycle theory is the cycle linking the two variables. In this work we focused on establishing the presence of each of the mutual effects, and have therefore analysed each of the two sides of the postulated cycle separately. This is a necessary first step, laying the ground for further research aimed at understanding the cycle dynamics and their involvement in mental health and disorders.

Indeed, models including "loopy", rather than linear connections between components have recently been the topic of several investigations. These illustrate the richness of the

observable behaviours in such systems, and the way subtle modulations of their dynamics can produce important large scale effects [3, 32, 58]. [58] investigated interactions between emotional states (happy vs sad stated induced by winning vs losing in a wheel of fortune draw) and learning about the rewarding properties of various slot machines in real human data and in simulations. Their results suggest that a positive-feedback effect of emotional state on the perception of outcomes may play a significant role in the emergence of mood instability. In a model of delusions, [32] show how small changes in "mood"- modelled as preference for one particular internal state- can, in conjunction with other factors (the quality of sensory information), dramatically alter behaviour, switching it from a relatively accurate pattern of inferences to one dominated by a "delusional" pattern of false and incorrigible inferences. In [32]'s case these effects were produced by reciprocal connections between mood, prior beliefs and actions: the agents' mood influenced their beliefs, which determined action choice; in the absence of good quality external information action choice was, in turn, used as a source of information, which resulted in the strengthening of initial beliefs and mood.

Such effects could also occur from the connection between attributions and beliefs. The simulations we presented in section **Simulations of an artificial agent** illustrate how reciprocal effects can enrich the space of dynamics in even a simple model of two interacting variables, amplifying randomly generated differences between agents, or producing latent vulnerabilities. These simulation only scratch the surface of a much larger space, as such systems can display high sensitivity to changes in inputs and parameters, as well as path dependencies, i.e., sensitivity to the ordering of inputs [3, 32]. There are a number of directions that need further exploration.

First, it is important to examine the parameter space systematically, i.e., to quantify the effects of exploring the ranges of individual parameters, as well as the effect of changes along multiple axes in the parameter space. Parameters of particular interest include attribution sensitivity to skill estimate level ($\beta$ parameters in our simulations), skill estimate sensitivity to attributions (differences between skill learning rates for internal vs external attributions in our simulations) and asymmetries between positive and negative outcomes (outcome-based learning rates in our simulation).

Going beyond examining parameters, one other particularly important direction is exploration of the space of models. This includes investigation of various linking functions between variables—e.g. in our simulations we chose *a priori* a sigmoid functional form of the dependencies of attributions on skill estimates—but also higher level dimensions of variability, such as time varying models, or higher order dependencies between variables. Such extensions would allow formulation and testing of hypotheses involving multiple time-scales of interaction or threshold-like mechanisms that might be responsible for catastrophic dynamics. As an example, one can imagine testing, in this context, whether an association between particularly low levels of beliefs about self and particularly strong effects of negative attributions on beliefs could model the onset of depression [2].

Finally, a crucial direction from a translational point of view is the investigation of perturbations and or environment manipulations that can produce qualitative shifts in the system dynamics, e.g. switches between "healthy" stable regimes and vicious cycles, or falls into of "delusional" [32] or other pathological attractor states [2, 3]. Harmful factors pushing the system away from equilibrium, and protective ones that prevent or dampen such dynamics [33, 34, 59, 60] would be particularly relevant for understanding the emergence of disorders and for developing preventive interventions or treatments. For instance, in connection with depression, simulations could be used to investigate differences between the effects of repeated small losses vs isolated exceptionally large losses, the effects of losses concentrated vs distributed in time (and their analogues for wins), and disproportionate effect of attributions for

particularly significant events. Better understanding of such path dependencies associated with negative outcomes and attributions might yield insights into targets for therapeutic interventions or mechanisms for maximising their effectiveness.

## Conclusion

To conclude, we propose a new framework for investigations into relationships between attributions and beliefs about self, and provide evidence of the dynamical nature of these variables and of reciprocal effects between them *at high temporal resolution*. We thereby validate one central tenet of the attribution self-representation cycle theory in healthy controls, helping bolster the important implications of this theory for the understanding of the emergence and maintenance of psychiatric disorders, as well as potentially for therapeutic approaches.

Data and code are available at https://osf.io/sdvf5/.

## Materials and methods

### Ethics statement

The experiment was conducted under ethics approval granted by Ethik-Kommission an der Medizinischen Fakultät der Eberhard-Karls-Universität und am Universitätsklinikum Tübingen and formal consent (on-line form) was obtained from all participants.

### Participants

Participants (self: $N = 122$, 78 female, 41 male, 3 missing gender data, aged 18–35 years, $M = 24.71$, $SD = 5.16$; other: $N = 92$, 58 female, 32 male, 2 missing gender data, aged 18–35 years, $M = 24.93$, $SD = 5.18$) were recruited on Prolific.

### Experiment timeline

Participants were directed to the experiment hosted on Gorilla. After providing informed consent, participants completed the three questionnaires. They were then presented with the detailed instructions for the task, and answered verification questions to check understanding of the task instructions. They then proceeded to the first session of the task. The second session in the "self" condition started with experimental trials directly and took part the following day. One week later, participants were provided with instructions for the other condition, which they they performed. Finally they performed the last session the following day. After the end of the last session participants were payed and provided with a feedback questions form, then with a separate question asking them to compare the two conditions. They were then fully debriefed.

### Attribution and skill estimate questions

Attributions were elicited with multiple-choice questions. After wins in the self condition, participants were asked "Why did you win the last trial? Pick the main cause:"; the response options were phrased as "simple maze", "few/ simple rotations", "luck", "my ability"; the corresponding version for the ability option in the other condition was "their ability". The question was adapted for losses by replacing "win" with "lose and for the other condition by replacing "you" with "they". Response options for losses were phrased as "complex maze", "many/difficult rotations", "bad luck", "my lack of skill", respectively; and "their lack of skill" for the other condition.

Participants were then asked "How good do you think you are at the task at this moment?" ("you" replaced by "they" in the other condition) and responded using a slider on a continuous scale with extremes labeled "very bad" and "very good".

## Model agnostic analyses

Model agnostic analyses of skill estimates were performed on skill estimate updates, computed as follows: skill estimates were z-scored within participant, and differences between two successive z-scored skill estimates were entered into analyses as skill estimate updates. For both model-agnostic and model-dependent analyses, attributions were relabelled as internal (self) vs external (maze complexity, rotations, luck). All tests were performed as permutation tests with 5000 samples, using the paired t-test statistic as the measure of interest and permuting labels within participant. E.g. to test for an effect of attribution conditioned on outcome = win, attribution labels (internal vs external) for all wins were permuted within participant; average update post internally attributed wins and average update post externally attributed wins were then computed for each participant; paired (int vs ext) t statistic was then computed across participants for each permutation. Two-sided p-values were computed across permutations.

Reported effect sizes are Hedge's corrected d

$$d = \left(1 - \frac{3}{4(l_1 + l_2) - 9}\right) \frac{\mu_1 - \mu_2}{\sqrt{\frac{(l_1-1)*\sigma_1^2 + (l_2-1)*\sigma_2^2}{l_1 + l_2 - 2}}},$$

where $\mu_1$ and $\mu_2$ are the sample means, $\sigma_1$, $\sigma_2$ are the sample standard deviations, and $l_1$, $l_2$ are the sample sizes.

Model agnostic analyses of attributions were performed on attribution proportions, computed as follows: for a given level of a factor of interest and a given attribution option, the summary statistic we used was the proportion of attributions to the option, out of all attributions provided for the factor level: e.g. for the effect of outcome on internal attributions in the self condition we compared the proportions of attributions to self out of all attributions provided for wins, vs the proportion of attributions to self out of all attributions provided for losses. We used quartile discretisation for factors of interest other than outcome, which were continuous. For each participant and each factor of interest, discretisation was performed on the z-scored factor values. Reported p-values are estimated with permutation tests, performed by permuting labels within participant.

All tests reported as significant survived Benjamini Hochberg ([61]) correction for multiple comparisons, unless otherwise stated (82 tests, $\alpha = 0.05$, highest p-value under threshold = 0.0464).

## Skill models

All skill models were variations of the basic Rescorla-Wagner model:

$$\delta_t = o_t - s_{t-1}$$

$$s_t = \begin{cases} s_{t-1} + \alpha * \delta_t & \text{if } t \neq t_0^{II} \\ s_{t-1} + \beta + \alpha * \delta_t & \text{otherwise, where} \end{cases}$$

$$t_0^{II} = \text{index of first trial of the second session.}$$

$$s_t = \text{underlying skill estimate at trial t}$$

$$o_t = \text{outcome of trial t}$$

$$\alpha = \text{learning rate}$$

$$\beta = \text{effect of break between sessions}$$

The reported skill estimate was assumed to be a noisy reading of the underlying skill estimate, drawn from a fixed Gaussian distribution $\mathcal{N}(0, 0.1)$. We refer to this as the baseline model. The rest of the models were obtained by allowing $\alpha$ to vary as follows:

- S: different learning rates for the first and second sessions (2 learing rates);

- O: different learning rates for wins and losses (2 learning rates);

- A: different learning rates for internal and external attributions plus a learning rate for outcomes with missing attributions (3 learning rates);

- SA: different learning rates for different attributions, separate for the two sessions (6 learning rates);

- SO: different learning rates for wins and losses, separate for the two sessions (4 learning rates);

- AO: different learning rates for different attributions, separately for wins and losses (6 learning rates);

- SAO: different learning rates for each combination of attribution and outcome, separately for the two sessions (12 learning rates).

Fitting and model comparison were performed separately for self and other. All models were coded in the pystan interface (https://pystan.readthedocs.io/en/latest/) to the STAN probabilistic programming language (https://mc-stan.org/), which we used to obtain samples from the posterior distribution over parameters and estimate model evidence (1000 iterations, 4 chains per model). All models were fitted in two versions: independently for each participant and as a hierarchical model over the entire population. For the hierarchical version, individual parameters were drawn from independent Beta distributions over each parameter at the population level:

$$
\begin{aligned}
\theta_p^i &\sim Beta(\alpha_i, \beta_i), \text{ where} \\
\alpha_i, \beta_i &= \text{population level parameters} \\
p &\quad \text{indexes participant} \\
i &\quad \text{indexes latent parameter.}
\end{aligned}
$$

## Attribution models

All attribution models were built on the same underlying structure:

$$
\begin{aligned}
s_{t,o} &= \boldsymbol{w}_o \cdot \boldsymbol{f}_t \forall o \in O \\
p_t(o) &= \frac{\exp(s_{t,o})}{\sum_{o \in O} \exp(s_{t,o})}, \text{ where} \\
s_{t,o} &= \text{score of response option o on trial t} \\
\boldsymbol{w}_o &= \text{feature weights for option o,} \\
&\quad \text{corresponding to the outcome on trial } t-1 \\
\boldsymbol{f}_t &= \text{feature values on trial t} \\
O &= \text{set of available response options} \\
p_t(o) &= \text{probability of choosing option o on trial t}
\end{aligned}
\tag{3}
$$

All models had separate parameters for wins and losses. Because participants have to choose one of the 4 available response options, models were equipped with independent parameters

for *three* of the options, and scores for the 4 options were constrained to sum to 0; thus parameters and preference for the Luck option were entirely derived from parameters and preferences for the other options and we do not report analyses of these parameters. Each model was defined by the features included:

- no features—baseline propensity for each attribution option;

- bias+ skill estimates (1 feature, 12 parameters per participant);

- bias+ performance features: proportion of correct key presses out of all key presses in trial; proportion of pauses out of all frames in trial (2 features, 18 parameters per participant);

- bias+ performance features (as above) and task features: length of correct path through maze, proportion of frames with maze in unusual (not UP) orientation in trial (4 features, 30 parameters per participant);

- skill estimates and task features (as above, 3 features, 24 parameters per participant);

- full model: skill estimates, performance features, task features (5 features, 36 parameters per participant).

Each model was fitted in two versions: independently for each participant and in a hierarchical version, assuming a Normal distribution at the population level for each parameter. Fitting and model comparison were performed separately for self and other. All models were coded in the pystan interface (https://pystan.readthedocs.io/en/latest/) to the STAN probabilistic programming language (https://mc-stan.org/), which we used to obtain samples from the posterior distribution over parameters and estimate model evidence (1000 iterations, 4 chains per model).

**Transforming feature weights into meaningful effects.** Because scores obtained from linear combinations of features are passed through a softmax to compute the probability of each attribution option, and because there are separate softmax transformations for wins and losses, direct comparison between feature weights for different options is not informative. As an example, the bias parameter for a given option does not directly and independently translate into the participant's preference for that option; it is the relationship between biases for the different options, entering into the softmax function, that determines preferences for the different options. In order to be able to meaningfully compare baseline preferences and the effects of various features we transformed these parameters as follows: for biases, we applied the softmax transformations to the bias parameters only, clamping all feature weights to 0, and compared the resulting probabilities. For a feature of interest $x$ we computed $xAo$, its contribution to choosing attribution option $A$ having encountered outcome $o$ (e. g sI+ for the contribution of skill estimates to making internal attributions for wins) as:

$$
\begin{aligned}
xAo &= \frac{1}{T}\sum_t \frac{\partial p_t(A)}{\partial x}\big|_{x=0}, \text{ where} \\
T &= \text{total number of trials} \\
p_t(A) &= \text{the probability of choosing attribution option A on trial t} \\
&= \frac{\exp(w_{Ao}\cdot f_t)}{\exp(w_{Io}\cdot f_t) + \exp(w_{Mo}\cdot f_t) + \exp(w_{Ro}\cdot f_t) + \exp(w_{Lo}\cdot f_t)}, \text{ where} \\
f_t &= \text{feature values on trial t} \\
w_{Xo} &= \text{weights for computing the score for attributions to X} \\
&\quad \text{after outcome o.}
\end{aligned}
\tag{4}
$$

Thus the derivative of $p_t(A)$, seen as a function of $x$, is evaluated at $x = 0$, and values are then averaged over all trials.

## Model comparison

Model comparison was performed with the WAIC score [62], an approximation for the out-of-samples predictive log density, computed as

$$
\widehat{WAIC} = \quad -\frac{1}{P}\sum\nolimits_{p=1}^{P} \log\left(\frac{1}{S}\Sigma_{s=1}^{S}\ p(X_p|\boldsymbol{\theta}_p^s)\right)
$$

$$
+\frac{1}{P}\sum\nolimits_{p=1}^{P}\left[\mathrm{Var}_{s=1}^{S}(\log\ p(X_p|\boldsymbol{\theta}_p^s))\right],
$$

(5)

where $\{\boldsymbol{\theta}_p^1, \boldsymbol{\theta}_p^2 \ldots \boldsymbol{\theta}_p^S\}$ is the set of samples from the posterior distribution over the vector of individual parameters, $\boldsymbol{\theta}_p$ for participant $p$.

In all cases WAIC score comparisons preferred hierarchical versions of the models; we therefore only report analyses of these. Mean posterior parameters for individual participants were entered into further analyses.

## Questionnaires

As detailed above, learning rates from the winning skill model varied along three axes: outcome, attribution and session. In analyses of relationships with questionnaire measures, parameters were averaged across the direction irrelevant for the analysis.

## Supporting information

**S1 Appendix. Staircase procedure.**
(DOCX)

**S2 Appendix. Empirical difficulty and skill.**
(DOCX)

**S1 Fig. Difficulty measure: Summary and sanity check.** Left: data pooled from all subjects; top: distribution of difficulty values; bottom: relationship between difficulty and the proportion of wins. Right: relationship between difficulty indifference point—difficulty value for which subject is equally likely to win or lose the trial—and the proportion of trials won out of all trials; each dot represents a subject; $r^2 = 0.6$, p-value = $3 * 10^{-7}$.
(TIF)

**S2 Fig. Accuracy for outcome prediction: Difficulty only vs difficulty and skill models.** Colours correspond to different $\mathbf{w}_d$ values in the skill and difficulty models; note that $\mathbf{w}_d = 0$ (purple) is equivalent to a model with skill only; black is used for the model with difficulty only. Top: overall accuracy; each dot represents one subject. Bottom: accuracy per difficulty level; mean ± s.e.m across subjects; difficulty was z-scored for each subject and discretised in 10-quantiles.
(TIF)

**S3 Fig. SAO skill model, quality of fit.** Example best (A) and worst (B) fit participants, SAO model of skill estimates, self condition. Top: participant responses vs underlying skill recovered with mean posterior parameters. Bottom: time series of participant responses and underlying skill recovered with mean posterior parameters.
(TIF)

**S4 Fig. SAO skill model parameters: Attribution response shuffling.** To test whether the attribution effects observed in model agnostic analyses were detectable in model parameters

we refitted the model to data with scrambled attributions and compared the observed differences in learning rates to the ones obtained in the real data. Attribution response shuffling: effect on difference between learning rates of the SAO model. Analyses were performed as follows: for each of 1000 permutations, attribution responses of each individual participant were shuffled and the SAO model was refitted. For each combination of outcome and session, the difference between the corresponding internal and external $\alpha$ parameters was averaged across participants. The resulting shuffle distribution is compared with the average difference obtained from fitting the real data. Left: self, right: other.
(TIF)

**S5 Fig. Model agnostic analyses of attributions: Other.** Features of interest and attributions summary other. Faded lines represents individual participants, bold lines represent mean ± s. e.m across participants. Orange: losses, teal: wins.
(TIF)

**S1 Table. SAO skill model parameters: Attribution response shuffling.**
(DOCX)

**S2 Table. Model parameters, winning attribution model: Effects of path length on internal attributions and attributions to Maze.**
(DOCX)

**S3 Table. Model parameters, winning attribution model: Effects of proportion of unusual orientations on internal attributions and attributions to Rotations.**
(DOCX)

**S4 Table. Model parameters, winning attribution model: Effects of reported skill on external attributions.**
(DOCX)

**S5 Table. Model parameters, winning attribution model: Effects of key press accuracy on external attributions.**
(DOCX)

## Author Contributions

**Conceptualization:** Elena Zamfir, Peter Dayan.

**Data curation:** Elena Zamfir.

**Formal analysis:** Elena Zamfir.

**Investigation:** Elena Zamfir, Peter Dayan.

**Methodology:** Elena Zamfir, Peter Dayan.

**Software:** Elena Zamfir.

**Validation:** Elena Zamfir, Peter Dayan.

**Visualization:** Elena Zamfir.

**Writing – original draft:** Elena Zamfir.

**Writing – review & editing:** Elena Zamfir, Peter Dayan.

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
