## [Decision Letter · Decision Letter 0]

5 Apr 2022

Dear Dr Zamfir, 

Thank you very much for submitting your manuscript "Interactions between attributions and beliefs at high temporal resolution : evidence from a novel task" for consideration at PLOS Computational Biology.

As with all papers reviewed by the journal, your manuscript was reviewed by members of the editorial board and by several independent reviewers. In light of the reviews (below this email), we would like to invite the resubmission of a significantly-revised version that takes into account the reviewers' comments.

While we ask you to address in full all the reviewers’s comments ("major revisions"), we encourage you to put extra effort in clarifying potential design flaws concerning skill judgment attribution and the relation between skills and outcomes; to better acknowledge previous literature about reinforcement learning biases (and their optimality), which does not seem to be exhaustively referenced and discussed in the current version; and to improve model and result presentation. The paper will be sent back to the original reviewers.

We cannot make any decision about publication until we have seen the revised manuscript and your response to the reviewers' comments. Your revised manuscript is also likely to be sent to reviewers for further evaluation.

Sincerely,

Stefano Palminteri

Associate Editor

PLOS Computational Biology

Samuel Gershman

Deputy Editor

PLOS Computational Biology

Reviewer's Responses to Questions

**Comments to the Authors:**

Reviewer #1: In this paper, Zamfir & Dayan use a novel sensorimotor skill learning task to test a computational model that posits a dynamic, cyclical relationship between outcome attributions (i.e., to the self or to external perturbations) and one’s belief about their abilities in a task. Specifically, their model predicts that individuals will produce greater updates to their belief about their skill after attributing an outcome internally; moreover, as skill improves over time (or rather, one’s belief about it), one will also be more likely to attribute positive outcomes internally, and negative outcomes externally. Using simulations, the authors show how their straightforward model can generate these effects. Then, using a novel navigational button-pressing task, they show that people’s task outcome attributions and ongoing skill judgments follow the predicted trends. Taken together, the results support their formalization of the so-called “attribution-self representation cycle theory.”

Overall this is a strong paper. The authors propose a novel formalization of an important idea, and introduce an impressive task to test their computational hypotheses. The modeling framework is rather straightforward and clear, and the simulation, fitting, and behavioral results generally provide strong support for the stated conclusions. I have several critiques concerning design and interpretation, highlighted below.

1) I may have missed something, but the implementation of the attribution and skill judgments seems to have a potential design flaw - the latter always followed the former during the task. It is possible that this could lead to a demand effect, where making a particular type of attribution biases one’s subsequent skill judgment. (This, in effect, might confound the results in the predicted direction I believe?) At the very least this potential drawback should be discussed, though perhaps additional analyses (or experiments with counterbalanced judgment orderings) could be implemented to directly address it.

2) Throughout the text, “skill” itself (objective performance) and “skill belief” are often conflated. This delineation could be made more explicit - what can we learn about discrepancies between these two constructs? Moreover, the fact that including both the objective and subjective skill metrics appears to be important for modeling attributions (if I followed the second fitting analysis correctly) is an interesting result that could be more thoroughly discussed.

3) The decision to include the “Other” condition is somewhat under-motivated. While there was at least one interesting result from that condition (the valence effects), how it is situated in the border manuscript felt a bit confusing. What were the explicit hypotheses about this condition?

4) The number of presses (accuracy) is used as the main performance metric. I think it could be useful to analyze, or at least mention, changes in response (or completion) time over different conditions and amounts of training. Or perhaps a “speed-accuracy tradeoff” metric could be computed? Might this represent a more direct measure of skill in this type of speeded sensorimotor task?

Little things:

-Significant versus non-significant correlations should be made clear in figure 9

-Typo “cuuld” on page 3

-In the figures, “Skill” might be changed to “Skill belief” or something similar

Reviewer #2: In this work, causal attributions were tested in online participants using an original skill game. Participants’ beliefs about their skills and causal attributions of experienced outcomes were collected trial by trial. The authors found that participants updated their skills beliefs more after internally than externally attributed outcomes; they also found that participants gave themselves more credit for wins and less blame for losses as their skills increased. Interestingly, these patterns correlated with questionnaire-based measures of self-esteem, locus of control and attributional style.

I particularly liked the computational part of the paper: the authors designed an elegant model in which the update of one’s skill belief is determined by a learning rate whose value depends on both the valence of the outcome and the participant's attribution (internal/external), allowing for fine-grained (trial by trial) tracking of potential interactions between skill beliefs and attributions. Model comparison revealed that allowing learning rates to differ for internal and external attributions improves the model evidence; both model-dependent and model-agnostic analyses showed a significant effect of attribution on skill estimates (internally attributed outcomes => stronger effects on beliefs about skills). I was less impressed with the correlations between the questionnaire-based measures and the individual parameters of the winning model, but given the usual low predictability of these questionnaires, the results presented are nonetheless far from negligible.

I have a few comments about the interpretation/discussion of the results:

1. I find the functional interpretation of your results a bit short, and particularly focused on personal-level dispositions (self-esteem, well-being, etc.). I think the (brief) discussion you make of your results would benefit from a discussion of other work close to yours, such as the studies conducted on learning asymmetries in the RL domain (e.g. Dorfman et al. 2019, Psych Sci, on self-serving bias in causal inference under benevolent vs adversarial conditions, or Chambon et al. 2021, Nat Hum Beh, on learning asymmetries in controllable vs uncontrollable environments, and perhaps also Cockburn et al. 2014, Neuron). These studies do not focus on skill sensitivity as in the present study, but rather on sensitivity to the controllability of the current situation, which somehow echoes low and high skill situations.

Interestingly, most of these results found a higher learning rate for positive outcomes, which is in stark contrast to what the you found here. I think this should be discussed further in the paper, especially since these authors suggest that a higher sensitivity to positive outcomes is not only a matter of maintaining a positive view of oneself (e.g. promoting self-esteem and well-being), but actually resolves a problem of credit assignment (e.g. Cockburn et al.) and/or allows for extra-reinforcement of actions that meet internal needs (e.g. Chambon et al.).

2. “Heightened attention to negative feedback can be essential for survival in harsh environments, and might be particularly relevant in learning contexts where it can be used to improve performance” : this claim is actually controversial at best (see Gershman 2015, PBR, or Chambon et al. 2021, NHB, for lack of learning rate adaptation as a function of the harshness of the environment), and contradicts optimality analyses from Cazé and van der Meer 2013 (cited in the present paper) which suggest that it is more advantageous to have an positivity bias in a harsh and poorly rewarded environment - which makes sense since positive outcomes are rarer in such environments and therefore more informative.

Also, given that your task does not vary the harshness of the environment, or, say, the amount of reward available in distinct blocks (e.g. “poor” and “rich” blocks), I find it difficult to conclude anything about the valence asymmetry found here (negative LR > positive LR).

3. Since psychiatric disorders are mentioned in the paper: it may be important to recall that abnormal control beliefs/causal inferences are not of one type and may vary along a continuum from the experience of increased internal control (e.g. delusion of omnipotence) to little or no control (e.g. delusion of control).

Thus, it might be relevant to specify in the paper how the parameters of the model (either competence level sensitivity, attribution sensitivity, or learning rate asymmetries) can be abnormally tuned to predict which end of this continuum (from delusion of omnipotence to delusion of control) the patient is on.

With respect to depression, your hypothesis of a "disproportionate effect of attributions for particularly significant events" is interesting but seems at odds with so-called "depressive realism" where, precisely, depression is explained by the absence of bias in attribution styles (no valence-induced asymmetries - i.e. no abnormal weighing of positive or negative events) or, more parsimoniously, by the patient’s passivity (depressed patients produce fewer actions than control participants, and are therefore less likely to make illusory causal attributions, see Matute et al., 2015, Front in Psychology).

Minor:

The paper does not seem to make a (strong) distinction between attribution style and locus of control (the latter being even used as a measure for the former). I think treating them as interchangeable constructs is questionable – and indeed debated in the literature. Locus measures expectations (or anticipations) about the future, while attributional style measures explanations about past outcomes.

Reviewer #3: The authors adapt classic reinforcement learning models to capture how agents update beliefs about their skills after receiving wins or losses. Thereby, the authors offer a new computational perspective on the long-debated question of how humans attribute wins and losses to internal vs. external causes. Overall, I find the article interesting and sound (i.e., the new models, the task, the analyses, and the cited literature). Apart from one suggestion regarding the relation between skill and outcomes, my main concern is that the presentation of the model, the results, and their implications is often rather convoluted. Please find my specific suggestions below.

1. Objective relation between skill (or difficulty) and outcomes: The authors mention this point in the discussion (in 3.2) but in my view this is too late and not completely justified. When reading the formulas for the skill belief updates (in 2.1), I immediately wondered how the model captures that actual skill is usually related to outcomes. Depending on the task, this relationship can vary, i.e., the probability of a win given a certain level of skill depends on how controllable the outcome is (e.g., in some tasks the outcome is completely determined by skill; in other task a positive outcome is still rather unlikely even at the highest level of skill). Crucially, agents can have undue assumptions about (i) their level of skill (which the newly introduced model captures) and/or about (ii) how much their skill determines the outcome.

a. I am wondering if the latter point could be formalized as p(o|s) and varied in the simulations of the model. Agents could form (and update) beliefs about this probability, which could then influence how they attribute, i.e., the p(a|o,s). Thereby, one could simulate tasks with different skill-outcome-relationships.

b. Even in the empirical data, p(o|s) could be obtained from the step-wise procedure that was used to adjust the difficulty of the employed task.

c. Even if such further analyses not possible in the current settings (as the authors imply in the discussion), it would have been really helpful for me to read early-on about these limitations in the scope of the model.

2. Presentation: Suggestions for more clarity. I’ll start with two suggestions that are more relevant in terms of content. After that, I have more nitty-gritty suggestions.

a. Simulations: I like that the authors first show simulations of their model and vary parameters. However, the motivation for varying specific components is not clearly given.

i. Often the authors mention verbally which “component” (e.g., “reciprocal influences,” “vulnerability,” etc.) is varied but they do not explicitly say to which parameter this component is linked (instead the readers needs to look into the precise values in the brackets).

ii. I had the impression that the authors want to show that initial conditions do not matter that much. So, many simulations are motivated by such very important checks. However, in my view many simulations with varying parameters could be motivated by showing how the model can capture different strategies in diverse populations. I think that overall a lot more could be done to illustrate how simulated agents with different learning rates (which capture levels of self-esteem, etc.) would learn. I am deliberately a bit vague here but I am convinced that the authors could show more simulations that give a better picture of what to expect from “simulated psychiatric patients.”

b. Higher learning rate for internal attributions after losses (Figure 7b): Seeing this, I first had the impression that participants were not self-serving. The authors discuss this point also in relation to the study by Müller-Pinzler et al. 2019). To understand this crucial point better, it would be helpful to show that the number of these trials is rather small because participants themselves make the “attribution decision.” For me, this result nicely exemplifies why it is so important to look at the interaction of outcomes and attributions. Therefore, it could be explained in more detail.

c. Title: “evidence from a novel task” This wording in the title is rather unspecific. Also, the wording “high temporal resolution” suggest an M/EEG study or some study using a specific measurement; maybe “trial-by-trial” or “step-by-step” would be better.

d. Abstract and introduction: I completely understand that the authors aim to apply their approach for understanding psychiatric disorders. But they often set the undue expectation that they will report data from patients. In the abstract, for example, it is unclear if the authors also assessed clinical scores. The “computational psychiatry aspect” could simply be mentioned as an outlook for future studies.

e. Abstract: “Substantial population of online participants:” The authors could directly mention the sample size in the abstract.

f. Introduction: The authors provide a really nice and detailed example on the fundamental attribution error. The authors refer to the substantial and long-established literature on attributions. There’s one bit that I didn’t quite get: How do the biases relate to the homeostatic regime?

g. Relationship with questionnaires: Here, the authors list some specific hypotheses but do not motivate them properly.

h. Figures:

i. Please do not use “See text for details”

ii. The y-axes is missing for Figures 8c and 8d.

iii. The models in Figure 7c have to be spelled out (and explained) in the caption.

iv. Often it is unclear how the data were split. E.g., quartiles in Figure 8a: Quartiles on what?

v. Figures 5a and 6c: These numbers add up to 1, don’t they? So, some forms of cumulative plots would be easier to grasp.

**Have the authors made all data and (if applicable) computational code underlying the findings in their manuscript fully available?**

Reviewer #1: Yes

Reviewer #2: Yes

Reviewer #3: **No: **The current version does not include links to the code.

PLOS authors have the option to publish the peer review history of their article (what does this mean?). If published, this will include your full peer review and any attached files.

Reviewer #1: No

Reviewer #2: No

Reviewer #3: No
---

## [Decision Letter · Decision Letter 1]

28 Aug 2022

Dear Dr Zamfir,

We are pleased to inform you that your manuscript 'Interactions between attributions and beliefs at trial-by-trial level : evidence from a novel computer game task' has been provisionally accepted for publication in PLOS Computational Biology.

Best regards,

Stefano Palminteri

Academic Editor

PLOS Computational Biology

Samuel Gershman

Section Editor

PLOS Computational Biology

Reviewer's Responses to Questions

**Comments to the Authors:**

Reviewer #1: The authors have done an excellent job responding to my comments in their revision.

Reviewer #2: The authors have addressed my concerns.

Reviewer #3: Thank you for addressing all my comments.

**Have the authors made all data and (if applicable) computational code underlying the findings in their manuscript fully available?**

Reviewer #1: Yes

Reviewer #2: Yes

Reviewer #3: Yes

PLOS authors have the option to publish the peer review history of their article (what does this mean?). If published, this will include your full peer review and any attached files.

Reviewer #1: No

Reviewer #2: No

Reviewer #3: No

---

## [Editor Report · Acceptance letter]

22 Sep 2022

PCOMPBIOL-D-22-00222R1 

Interactions between attributions and beliefs at trial-by-trial level : evidence from a novel computer game task

Dear Dr Zamfir,

I am pleased to inform you that your manuscript has been formally accepted for publication in PLOS Computational Biology. Your manuscript is now with our production department and you will be notified of the publication date in due course.

With kind regards,

Agnes Pap
